# A New Perspective on Pool-Based Active Classification and False-Discovery Control

**Lalit Jain, Kevin Jamieson**
{lalitj, jamieson}@cs.washington.edu
Paul G. Allen School of Computer Science & Engineering
University of Washington, Seattle, WA

## Abstract

In many scientific settings there is a need for adaptive experimental design to guide the process of identifying regions of the search space that contain as many true positives as possible subject to a low rate of false discoveries (i.e. false alarms). Such regions of the search space could differ drastically from a predicted set that minimizes 0/1 error and accurate identification could require very different sampling strategies. Like active learning for binary classification, this experimental design cannot be optimally chosen a priori, but rather the data must be taken sequentially and adaptively. However, unlike classification with 0/1 error, collecting data adaptively to find a set with high true positive rate and low false discovery rate (FDR) is not as well understood. In this paper we provide the first provably sample efficient adaptive algorithm for this problem. Along the way we highlight connections between classification, combinatorial bandits, and FDR control making contributions to each.

## 1   Introduction

As machine learning has become ubiquitous in the biological, chemical, and material sciences, it has become irresistible to use these techniques not only for making inferences about *previously* collected data, but also for guiding the data collection process, closing the loop on inference and data collection [10, 38, 41, 39, 33, 31]. However, though collecting data randomly or non-adaptively can be inefficient, ill-informed ways of collecting data adaptively can be catastrophic: a procedure could collect some data, adopt an incorrect belief, collect more data based on this belief, and leave the practitioner with insufficient data in the right places to infer anything with confidence.

In a recent high-throughput protein synthesis experiment [33], thousands of short amino acid sequences (length less than 60) were evaluated with the goal of identifying and characterizing a subset of the pool of all possible sequences ($\approx 10^{80}$) containing many sequences that will fold into stable proteins. That is, given an evaluation budget that is just a minuscule proportion of the total number of sequences, the researchers sought to make predictions about individual sequences that would never be evaluated. An initial first round of sequences uniformly sampled from a predefined subset were synthesized to observe whether each sequence was in the set of sequences that will fold, $\mathcal{H}_1$, or in $\mathcal{H}_0 = \mathcal{H}_1^c$. Treating this as a classification problem, a linear logistic regression classifier was trained, using these labels and physics based features. Then a set of sequences to test in the next round were chosen to maximize the probability of folding according to this empirical model - a procedure repeated twice more. This strategy suffers two flaws. First, selecting a set to maximize the likelihood of hits given past rounds' data is effectively using logistic regression to perform *optimization* similar to follow-the-leader strategies [14]. While more of the sequences *evaluated* may fold, these observations may provide little information about whether sequences that were *not* evaluated will fold or not. Second, while it is natural to employ logistic regression or the SVM

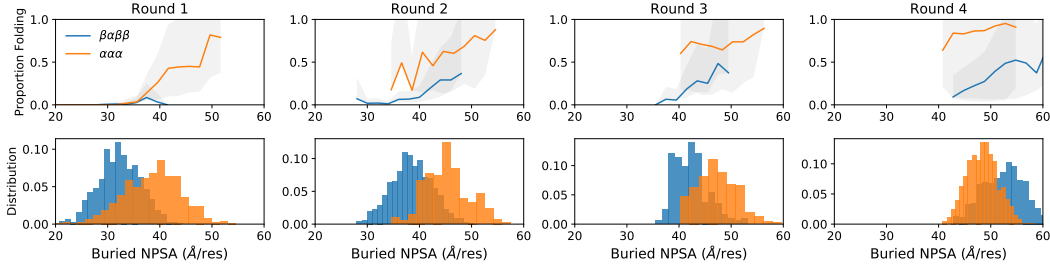

Figure 1: The distribution of a feature that is highly correlated with the fitted logistic model (bottom plot) and the proportion of sequences that fold (top plot). The distribution of this feature for the sequences drifts right.

to discriminate between binary outcomes (e.g., fold/not-fold), in many scientific applications the property of interest is incredibly rare and an optimal classifier will just predict a single class e.g. not fold. This is not only an undesirable inference for prediction, but a useless signal for collecting data to identify those regions with higher, but still unlikely, probabilities of folding. Consider the data of [33] reproduced in Figure 1, where the proportion of sequences that fold along with their distributions for a particularly informative feature (Buried NPSA) are shown in each round for two different protein topologies (notated $\beta\alpha\beta\beta$ and $\alpha\alpha\alpha$). In the last column of Figure 1, even though most of the sequences evaluated are likely to fold, we are sampling in a small part of the overall search space. This limits our overall ability to identify under-explored regions that could potentially contain many sequences that fold, even though the logistic model does not achieve its maximum there. On the other hand, in the top plot of Figure 1, sequences with topology $\beta\alpha\beta\beta$ (shown in blue) so rarely folded that a near-optimal classifier would predict "not fold" for every sequence.

Instead of using a procedure that seeks to maximize the probability of folding or classifying sequences as fold or not-fold, a more natural objective is to predict a set of sequences $\pi$ in such a way as to *maximize the true positive rate (TPR)* $|\mathcal{H}_1 \cap \pi|/|\mathcal{H}_1|$ while *minimizing the false discovery rate (FDR)* i.e. $|\mathcal{H}_0 \cap \pi|/|\pi|$. That is, $\pi$ is chosen to contain a large number of sequences that fold while the proportion of false-alarms among those predicted is relatively small. For example, if a set $\pi$ for $\beta\alpha\beta\beta$ was found that maximized TPR subject to FDR being less than $9/10$ then $\pi$ would be non-empty with the guarantee that at least one in every 10 suggestions was a true-positive; not ideal, but making the best of a bad situation. In some settings, such as for topology $\alpha\alpha\alpha$ (shown in orange), training a classifier to minimize 0/1 loss may be reasonable. Of course, before seeing any data we would not know whether classification is a good objective so it is far more conservative to optimize for maximizing the number of discoveries.

**Contributions.** We propose the first provably sample-efficient adaptive sampling algorithm for maximizing TPR subject to an FDR constraint. This problem has deep connections to active binary classification (e.g., active learning) and pure-exploration for combinatorial bandits that are necessary steps towards motivating our algorithm. We make the following contributions:

1. We improve upon state of the art sample complexity for pool-based active classification in the agnostic setting providing novel sample complexity bounds that do not depend on the disagreement-coefficient for sampling with or without replacement. Our bounds are more granular than previous results as they describe the contribution of a single example to the overall sample complexity.
2. We highlight an important connection between active classification and combinatorial bandits. Our results follow directly from our improvements to the state of the art in combinatorial bandits, extending methods to be near-optimal for classes that go beyond matroids where one need not sample every arm at least once.
3. Our main contribution is the development and analysis of an adaptive sampling algorithm that minimizes the number of samples to identify the set that maximizes the true positive rate subject to a false discovery constraint. To the best of our knowledge, this is the first work to demonstrate a sample complexity for this problem that is provably better than non-adaptive sampling.

## 1.1 Pool Based Classification and FDR Control

Here we describe what is known as the pool-based setting for active learning with stochastic labels. Throughout the following we assume access to a finite set of items $[n] = \{1, \cdots, n\}$ with an associated label space $\{0, 1\}$. The items can be fixed vectors $\{x_i\}_{i=1}^n \in \mathbb{R}^d$ but we do not restrict

to this case. Associated to each $i \in [n]$ there is a Bernoulli distribution $\text{Ber}(\eta_i)$ with $\eta_i \in [0, 1]$. We imagine a setting where in each round a player chooses $I_t \in [n]$ and observes an independent random variable $Y_{I_t, t}$. For any $i$, $Y_{i,t} \sim \text{Ber}(\eta_i)$ are i.i.d. Borrowing from the multi-armed bandit literature, we may also refer to the items as *arms*, and *pulling an arm* is receiving a sample from its corresponding label distribution. We will refer to this level of generality as the **stochastic noise** setting. The case when $\eta_i \in \{0, 1\}$, i.e. each point $i \in [n]$ has a deterministic label $Y_{i,j} = \eta_i$ for all $j \geq 1$, will be referred to as the **persistent noise** setting. In this setting we can define $\mathcal{H}_1 = \{i : \eta_i = 1\}, \mathcal{H}_0 = [n] \setminus \mathcal{H}_1$. This is a natural setting if the experimental noise is negligible so that performing the same measurement multiple times gives the same result. A classifier is a decision rule $f : [n] \to \{0, 1\}$ that assigns each item $i \in [n]$ a fixed label. We can identify any such decision rule with the set of items it maps to 1, i.e. the set $\pi = \{i : i \in [n], f(i) = 1\}$. Instead of considering all possible sets $\pi \subset [n]$, we will restrict ourselves to a smaller class $\Pi \subset 2^{[n]}$. With this interpretation, one can imagine $\Pi$ being a combinatorial class, such as the collection of all subsets of $[n]$ of size $k$, or if we have features, $\Pi$ could be the sets induced by the set of all linear separators over $\{x_i\}$.

The *classification error*, or *risk* of a classifier is given by the expected number of incorrect labels, i.e.

$$R(\pi) = \mathbb{P}_{i \sim \text{Unif}([n]), Y_i \sim \text{Ber}(\eta_i)} (\pi(i) \neq Y_i) = \frac{1}{n}(\sum_{i \notin \pi} \eta_i + \sum_{i \in \pi}(1 - \eta_i))$$

for any $\pi \in \Pi$. In the case of persistent noise the above reduces to $R(\pi) = \frac{|\pi \cap \mathcal{H}_0| + |\pi^c \cap \mathcal{H}_1|}{n} = \frac{|\mathcal{H}_1 \Delta \pi|}{n}$ where $A \Delta B = (A \cup B) - (A \cap B)$ for any sets $A, B$.

**Problem 1:(Classification)** Given a hypothesis class $\Pi \subseteq 2^{[n]}$ identify $\pi^* := \operatorname*{argmin}_{\pi \in \Pi} R(\pi)$ by requesting as few labels as possible.

As described in the introduction, in many situations we are not interested in finding the lowest risk classifier, but instead returning $\pi \in \Pi$ that contains many *discoveries* $\pi \cap \mathcal{H}_1$ without too many false alarms $\pi \cap \mathcal{H}_0$. Define $\eta_\pi := \sum_{i \in \pi} \eta_x$. The *false discovery rate (FDR)* and *true positive rate (TPR)* of a set $\pi$ in the stochastic noise setting are given by

$$FDR(\pi) := 1 - \frac{\eta_\pi}{|\pi|} \quad \text{and} \quad TPR(\pi) := \frac{\eta_\pi}{\eta_{[n]}}$$

In the case of persistent noise, $FDR(\pi) = \frac{|\mathcal{H}_0 \cap \pi|}{|\pi|} = 1 - \frac{|\mathcal{H}_1 \cap \pi|}{|\pi|}$ and $TPR(\pi) = \frac{|\mathcal{H}_1 \cap \pi|}{|\mathcal{H}_1|}$. A convenient quantity that we can use to reparametrize these quantities is the *true positives: $TP(\pi) := \sum_{i \in \pi} \eta_i$*. Throughout the following we let $\Pi_\alpha = \{\pi \in \Pi : FDR(\pi) \leq \alpha\}$.

**Problem 2:(Combinatorial FDR Control)** Given an $\alpha \in (0, 1)$ and hypothesis class $\Pi \subseteq 2^{[n]}$ identify $\pi_\alpha^* = \operatorname*{argmax}_{\pi \in \Pi, FDR(\pi) \leq \alpha} TPR(\pi)$ by requesting as few labels as possible.

In this work we are *agnostic* about how $\eta$ relates to $\Pi$, ala [2, 20]. For instance we do *not* assume the Bayes classifier, $\operatorname*{argmin}_{B \in \{0,1\}^n} R(B)$ is contained in $\Pi$.

## 2   Related Work

**Active Classification.** Active learning for binary classification is a mature field (see surveys [36, 25] and references therein). The major theoretical results of the field can coarsely be partitioned into the streaming setting [2, 6, 20, 26] and the pool-based setting [19, 24, 32], noting that algorithms for the former can be used for the latter, [2], an inspiration for our algorithm, is such an example. These results rely on different complexity measures known as the splitting index, the teaching dimension, and (arguably the most popular) the disagreement coefficient.

**Computational Considerations.** While there have been remarkable efforts to make some of these methods more computationally efficient [6, 26], we believe even given infinite computation, many of these previous works are fundamentally inefficient from a sample complexity perspective. This stems from the fact that when applied to common combinatorial classes (for example the collection of all subsets of size $k$), these algorithms have sample complexities that are off by at least $\log(n)$ factors from the best algorithms for these classes. Consequently, in our work we focus on sample complexity alone, and leave matters of computational efficiency for future work.

**Other Measures.** Given a static dataset, the problem of finding a set or classifier that maximizes TPR subject to FDR-control in the information retrieval community is also known as finding a binary classifier that maximizes recall for a given precision level. There is extensive work on the non-adaptive sample complexity of computing measures related to precision and recall such as AUC, and F-scores [35, 9, 1]. However, there have been just a few works that consider adaptively collecting data with the goal of maximizing recall with precision constraints [34, 5], with the latter work being the most related. We will discuss it further after the statement of our main result. In [34], the problem of adaptively estimating the whole ROC curve for a threshold class is considered under a monotonicity assumption on the true positives; our algorithm is agnostic to this assumption.

**Combinatorial Bandits:** The pure-exploration combinatorial bandit game has been studied for the case of all subsets of $[n]$ of size $k$ known as the Top-K problem [22, 29, 30, 28, 37, 17], the bases of a rank-$k$ matroid (for which Top-K is a particular instance) [18, 23, 15], and in the general case [11, 16]. The combinatorial bandit component of our work (see Section 3.2) is closest to [11]. The algorithm of [11] uses a disagreement-based algorithm in the spirit of Successive Elimination for bandits [22], or the $A^2$ for binary classification [2]. Exploring precisely what $\log$ factors are necessary has been an active area. [16] demonstrates a family of instances in which they show in the worst-case, the sample complexity must scale with $\log(|\Pi|)$. However, there are many classes like best-arm identification and matroids where sample complexity does *not* scale with $\log(|\Pi|)$ (see references above). Our own work provides some insight into what $\log$ factors are necessary by presenting our results in terms of VC dimension. In addition, we discuss situations when a $\log(n)$ could potentially be avoided by appealing to Sauer's lemma in the supplementary material.

**Multiple Hypothesis Testing.** Finally, though this work shares language with the adaptive multiple-hypothesis testing literature [12, 27, 42, 40], the goals are different. In that setting, there is a set of $n$ hypothesis tests, where the null is that the mean of each distribution is zero and the alternative is that it is nonzero. [27] designs a procedure that adaptively allocates samples and uses the Benjamini-Hochberg procedure [4] on $p$-values to return an FDR-controlled set. We are not generally interested in finding which individual arms have means that are above a fixed threshold, but instead, given a hypothesis class we want to return an FDR controlled set in the hypothesis class with high TPR. This is the situation in many structured problems in scientific discovery where the set of arms corresponds to an extremely large set of experiments and we have feature vector associated with each arm. We can't run each one but we may have some hope of identifying a region of the search space which contains many discoveries. In summary, unlike the setting of [27], $\Pi$ encodes structure among the sets, we do not insist each item is sampled, and we are allowing for persistent labels - overall we are solving a different and novel problem.

# 3 Pool Based Active Classification

We first establish a pool based active classification algorithm that motivates our development of an adaptive algorithm for FDR-control. For each $i$ define $\mu_i := 2\eta_i - 1 \in [-1, 1]$ so $\eta_i = \frac{1+\mu_i}{2}$. By a simple manipulation of the definition of $R(\pi)$ above we have

$$R(\pi) = \frac{1}{n}\sum_{i=1}^{n}\eta_i + \frac{1}{n}\sum_{i\in\pi}(2\eta_i - 1) = \frac{1}{n}\sum_{i=1}^{n}\eta_i - \frac{1}{n}\sum_{i\in\pi}\mu_i$$

so that $\operatorname*{argmin}_{\pi\in\Pi} R(\pi) = \operatorname*{argmax}_{\pi\in\Pi} \sum_{i\in\pi}\mu_i$. Define $\mu_\pi := \sum_{i\in\pi}\mu_i$. If for some $i \in [n]$ we map the $j$th draw of its label $Y_{i,j} \mapsto 2Y_{i,j} - 1$, then $\mathbb{E}[2Y_{i,j} - 1] = \mu_i$ and returning an optimal classifier in the set is equivalent to returning $\pi \in \Pi$ with the largest $\mu_\pi$. Algorithm 1 exploits this.

The algorithm maintains a collection of active sets $\mathcal{A}_k \subseteq \Pi$ and an active set of items $T_k \subseteq [n]$ which is the symmetric difference of all sets in $\mathcal{A}_k$. To see why we only sample in $T_k$, if $i \in \cap_{\pi\in\mathcal{A}_k}\pi$ then $\pi$ and $\pi'$ agree on the label of item $i$, and any contribution of arm $i$ is canceled in each difference $\widehat{\mu}_\pi - \widehat{\mu}_{\pi'} = \widehat{\mu}_{\pi\setminus\pi'} - \widehat{\mu}_{\pi'\setminus\pi}$ for all $\pi, \pi' \in \mathcal{A}_k$ so we should not pay to sample it. In each round sets $\pi$ with lower empirical means that fall outside of the confidence interval of sets with higher empirical means are removed. There may be some concern that samples from previous rounds are reused. The estimator $\widehat{\mu}_{\pi',k} - \widehat{\mu}_{\pi,k} = \frac{n}{t}\sum_{s=1}^{t} R_{I_t,s}(\mathbf{1}(I_s \in \pi'\setminus\pi) - \mathbf{1}(I_s \in \pi\setminus\pi'))$ depends on *all* $t$ samples up to the $t$-th round, each of which is uniformly and independently drawn at each step. Thus each summand is an unbiased estimate of $\mu_{\pi'} - \mu_\pi$. However, for $\pi, \pi'$ active in round $k$, as explained

above, a summand is only non-zero if $I_s \in \pi\Delta\pi' \subset T_k$ hence we only need to observe $R_{I_t,s}$ if $I_t \in T_k$ so the estimate of $\widehat{\mu}_{\pi',k} - \widehat{\mu}_{\pi,k}$ is unbiased.

In practice, since the number of samples that land in $T_k$ follow a binomial distribution, instead of using rejection sampling we could instead have drawn a single sample from a binomial distribution and sampled that many uniformly at random from $T_k$.

---

**Input**: $\delta, \Pi \subset 2^{[n]}$, Confidence bound $C(\pi', \pi, t, \delta)$.
Let $\mathcal{A}_1 = \Pi, T_1 = (\cup_{\pi \in \mathcal{A}_1}\pi) - (\cap_{\pi \in \mathcal{A}_1}\pi), k = 1, \mathcal{A}_k$ will be the active sets in round $k$
**for** $t = 1, 2, \cdots$
  **if** $t == 2^k$**:**
    Set $\delta_k = .5\delta/k^2$. For each $\pi, \pi'$ let
    $\widehat{\mu}_{\pi',k} - \widehat{\mu}_{\pi,k} = \frac{n}{t}(\sum_{s=1}^t R_{I_s,s}\mathbf{1}\{I_s \in \pi' \setminus \pi\} - \sum_{s=1}^t R_{I_s,s}\mathbf{1}\{I_s \in \pi \setminus \pi'\})$
    Set $\mathcal{A}_{k+1} = \mathcal{A}_k - \{\pi \in \mathcal{A}_k : \exists\pi' \in \mathcal{A}_k \text{with } \widehat{\mu}_{\pi',k} - \widehat{\mu}_{\pi,k} > C(\pi', \pi, t, \delta_k)\}$.
    Set $T_{k+1} = (\cup_{\pi \in \mathcal{A}_{k+1}}\pi) - (\cap_{\pi \in \mathcal{A}_{k+1}}\pi)$.
    $k \leftarrow k + 1$
  **endif**
  **Stochastic Noise:**
    If $T_k = \emptyset$, **Break**. Otherwise, draw $I_t$ uniformly at random from $[n]$ and if $I_t \in T_k$ receive an
    associated reward $R_{I_t,t} = 2Y_{I_t,t} - 1, Y_{I_t,t} \overset{iid}{\sim} \text{Ber}(\eta_{I_t})$.
  **Persistent Noise:**
    If $T_k = \emptyset$ or $t > n$, **Break**. Otherwise, draw $I_t$ uniformly at random from $[n] \setminus \{I_s : 1 \leq s < t\}$
    and if $I_t \in T_k$ receive associated reward $R_{I_t,t} = 2Y_{I_t,t} - 1, Y_{I_t,t} = \eta_{I_t}$.
**Output:** $\pi' \in \mathcal{A}_k$ such that $\widehat{\mu}_{\pi',k} - \widehat{\mu}_{\pi,k} \geq 0$ for all $\pi \in \mathcal{A}_k \setminus \pi'$

Algorithm 1: Action Elimination for Active Classification

For any $\mathcal{A} \subseteq 2^{[n]}$ define $V(\mathcal{A})$ as the VC-dimension of a collection of sets $\mathcal{A}$. Given a family of sets, $\Pi \subseteq 2^{[n]}$, define $B_1(k) := \{\pi \in \Pi : |\pi| = k\}$, $B_2(k, \pi') := \{\pi \in \Pi : |\pi\Delta\pi'| = k\}$. Also define the following complexity measures:

$$V_\pi := V(B_1(|\pi|)) \wedge |\pi| \quad \text{and} \quad V_{\pi,\pi'} := \max\{V(B_2(|\pi\Delta\pi'|, \pi)), V(B_2(|\pi\Delta\pi'|, \pi'))\} \wedge |\pi\Delta\pi'|$$

In general $V_\pi, V_{\pi,\pi'} \leq V(\Pi)$. A contribution of our work is the development of confidence intervals that do not depend on a union bound over the class but instead on local VC dimensions. These are described carefully in Lemma 1 in the supplementary materials.

**Theorem 1** *For each $i \in [n]$ let $\mu_i \in [-1, 1]$ be fixed but unknown and assume $\{R_{i,j}\}_{j=1}^\infty$ is an i.i.d sequence of random variables such that $\mathbb{E}[R_{i,j}] = \mu_i$ and $R_{i,j} \in [-1, 1]$. Define $\widetilde{\Delta}_\pi = |\mu_\pi - \mu_{\pi^*}|/|\pi\Delta\pi^*|$, and*

$$\tau_\pi = \frac{V_{\pi,\pi^*}}{|\pi^*\Delta\pi|}\frac{1}{\widetilde{\Delta}_\pi^2}\log\left(n\log(\widetilde{\Delta}_\pi^{-2})/\delta\right).$$

*Using $C(\pi, \pi', t, \delta) := \sqrt{\frac{8|\pi\Delta\pi'|nV_{\pi,\pi'}\log(\frac{n}{\delta})}{t}} + \frac{4nV_{\pi,\pi'}\log(\frac{n}{\delta})}{3t}$ for a fixed constant c, with probability greater than $1 - \delta$, in the **stochastic noise** setting Algorithm 1 returns $\pi_*$ after a number of samples no more than $c\sum_{i=1}^n \max_{\pi \in \Pi:i \in \pi\Delta\pi^*} \tau_\pi$ and in the **persistent noise** setting the number of samples needed is no more than $c\sum_{i=1}^n \min\{1, \max_{\pi \in \Pi:i \in \pi\Delta\pi^*} \tau_\pi\}$*

Heuristically, the expression $1/|\pi\Delta\pi^*|\widetilde{\Delta}_\pi^2$ roughly captures the number of times we would have to sample each $i \in \pi\Delta\pi^*$ to ensure that we can show $\mu_{\pi^*} > \mu_\pi$. Thus in the more general case, we may expect that we can stop pulling a specific $i$ each set $\pi$ such that $i \in \pi\Delta\pi^*$ is removed - accounting for the expression $\max_{\pi \in \Pi, i \in \pi\Delta\pi^*} \tau_\pi$. The VC-dimension and the logarithmic term in $\tau_\pi$ is discussed further below and primarily comes from a careful union bound over the class $\Pi$. One always has $1/|\pi^*\Delta\pi| \leq V_{\pi,\pi^*}/|\pi^*\Delta\pi| \leq 1$ and both bounds are achievable by different classes $\Pi$.

In addition, in terms of risk $\widetilde{\Delta}_\pi = |\mu_\pi - \mu_{\pi^*}|/|\pi\Delta\pi^*| = n|R(\pi) - R(\pi^*)|/|\pi\Delta\pi^*|$. Since sampling is done without replacement for persistent noise, there are improved confidence intervals that one can use in that setting described in Lemma 1 in the supplementary materials. Finally, if we had sampled non-adaptively, i.e. without rejection sampling, we would have had a sample complexity of $O(n\max_{i \in [n]} \max_{\pi:\Pi:i \in \pi\Delta\pi^*} \tau_\pi)$.

## 3.1 Comparison with previous Active Classification results.

**One Dimensional Thresholds:** In the bound of Theorem 1, a natural question to ask is whether the $\log(n)$ dependence can be improved. In the case of nested classes, such as thresholds on a line, we can replace the $\log(n)$ with a $\log\log(n)$ using empirical process theory. This leads to confidence intervals dependent on $\log\log(n)$ that can be used in place of $C(\pi', \pi, t, \delta)$ in Algorithm 1 (see sections C for the confidence intervals and 3.2 for a longer discussion). Under specific noise models we can give a more interpretable sample complexity. Let $h \in (0, 1]$, $\alpha \geq 0$, $z \in [0, 1]$ for some $i \in [n-1]$ and assume that $\eta_i = \frac{1}{2} + \frac{\text{sign}(z-i/n)}{2} h|z - i/n|^\alpha$ so that $\mu_i = h|z - i/n|^\alpha \text{sign}(z - i/n)$ (this would be a reasonable noise model for topology $\alpha\alpha\alpha$ in the introduction). Let $\Pi = \{[k] : k \leq n\}$. In this case, inspecting the dominating term of Theorem 1 for $i \in \pi^*$ we have $\arg\max_{\pi \in \Pi : i \in \pi \Delta \pi^*} \frac{V_{\pi,\pi^*}}{|\pi \Delta \pi^*|} \frac{1}{\bar{\Delta}_\pi^2} = [i]$ and takes a value of $\left(\frac{1+\alpha}{h}\right)^2 n^{-1}(z - i/n)^{-2\alpha-1}$. Upper bounding the other terms and summing, the sample complexities can be calculated to be $O(\log(n)\log(\log(n)/\delta)/h^2)$ if $\alpha = 0$, and $O(n^{2\alpha}\log(\log(n)/\delta)/h^2)$ if $\alpha > 0$. These rates match the minimax lower bound rates given in [13] up to $\log\log$ factors. Unlike the algorithms given there, our algorithm works in the *agnostic* setting, i.e. it is making no assumptions about whether the Bayes classifier is in the class. In the case of non-adaptive sampling, the sum is replaced with the max times $n$ yielding $n^{2\alpha+1}\log(\log(n)/\delta)/h^2$ which is substantially worse than adaptive sampling.

**Comparison to previous algorithms:** One of the foundational works on active learning is the DHM algorithm of [20] and the $A^2$ algorithm that preceded it [2]. Similar in spirit to our algorithm, DHM requests a label only when it is uncertain how $\pi^*$ would label the current point. In general the analysis of the DHM algorithm can not characterize the contribution of each arm to the overall sample complexity leading to sub-optimal sample complexity for combinatorial classes. For example in the the case when $\Pi = \{[i]\}_{i=1}^n$, with $i^* = \arg\max_{i \in [n]} \mu_i$, ignoring logarithmic factors, one can show for this problem the bound of Theorem 1 of [20] scales like $n^2 \max_{i \neq i_*}(\mu_{i^*} - \mu_i^{-2})$ which is substantially worse than our bound for this problem which scales like $\sum_{i \neq i_*} \Delta_i^{-2}$. Similar arguments can be made for other combinatorial classes such as all subsets of size $k$. While we are not particularly interested in applying algorithms like DHM to this specific problem, we note that the style of its analysis exposes such a gross inconsistency with past analyses of the best known algorithms that the approach leaves much to be desired. For more details, please see A.2 in the supplementary materials.

## 3.2 Connections to Combinatorial Bandits

A closely related problem to classification is the *pure-exploration combinatorial bandit* problem. As above we have access to a set of arms $[n]$, and associated to each arm is an unknown distribution $\nu_i$ with support in $[-1, 1]$ - which is arbitrary not just a Bernoulli label distribution. We let $\{R_{i,j}\}_{j=1}^\infty$ be a sequence of random variables where $R_{i,j} \sim \nu_i$ is the $j$th (i.i.d.) draw from $\nu_i$ satisfying $\mathbb{E}[R_{i,j}] = \mu_i \in [-1, 1]$. In the persistent noise setting we assume that $\nu_i$ is a point mass at $\mu_i \in [-1, 1]$. Given a collection of sets $\Pi \subseteq 2^{[n]}$, for each $\pi \in \Pi$ we define $\mu_\pi := \sum_{i \in \pi} \mu_i$ the sum of means in $\pi$. The pure-exploration for combinatorial bandit problem asks, given a hypothesis class $\Pi \subseteq 2^{[n]}$ identify $\pi^* = \arg\max_{\pi \in \Pi} \mu_\pi$ by requesting as few labels as possible. The combinatorial bandit extends many problems considered in the multi-armed bandit literature. For example setting $\Pi = \{\{i\} : i \in [n]\}$ is equivalent to the best-arm identification problem.

The discussion at the start of Section 3 shows that the classification problem can be mapped to combinatorial bandits - indeed minimizing the 0/1 loss is equivalent to maximizing $\mu_\pi$. In fact, Algorithm 1 gives state of the art results for the pure exploration combinatorial bandit problem and furthermore Theorem 1 holds verbatim. Algorithm 1 is similar to previous action elimination algorithms for combinatorial bandits in the literature, e.g. Algorithm 4 in [11]. However, unlike previous algorithms, we do not insist on sampling each item once, an unrealistic requirement for classification settings - indeed, not having this constraint allows us to reach minimax rates for classification in one dimensions as discussed above. In addition, this resolves a concern brought up in [11] for elimination being used for PAC-learning. We prove Theorem 1 in this more general setting in the supplementary materials, see A.3.

The connection between FDR control and combinatorial bandits is more direct: we are seeking to find $\pi \in \Pi$ with maximum $\eta_\pi$ subject to FDR-constraints. This already highlights a key difference

<div style="border:1px solid">

**Input:** Confidence bounds $C_1(\pi, t, \delta), C_2(\pi, \pi', t, \delta)$

$\mathcal{A}_k \subset \Pi$ will be the set of active sets in round $k$. $\mathcal{C}_k \subset \Pi$ is the set of FDR-controlled policies in round $k$.

$\mathcal{A}_1 = \Pi, \mathcal{C}_1 = \emptyset, S_1 = \cup_{\pi \in \Pi} \pi, T_1 = \bigcup_{\pi \in \Pi} \pi - \bigcap_{\pi \in \Pi} \pi, k = 1$.

**for** $t = 1, 2, \cdots$

  **if** $t = 2^k$:

    Let $\delta_k = .25\delta/k^2$

    For each set $\pi \in \mathcal{A}_k$, and each pair $\pi', \pi \in \mathcal{A}_k$ update the estimates:

    $\widehat{FDR}(\pi) := 1 - \frac{n}{|\pi| t} \sum_{s=1}^t Y_{I_s, s} \mathbf{1}\{I_s \in \pi\}$

    $\widehat{TP}(\pi') - \widehat{TP}(\pi) := \frac{n}{t} \left( \sum_{s=1}^t Y'_{J_s, s} \mathbf{1}\{J_s \in \pi' \backslash \pi\} - \sum_{s=1}^t Y'_{J_s, s} \mathbf{1}\{J_s \in \pi \backslash \pi'\} \right)$

    Set $\mathcal{C}_{k+1} = \mathcal{C}_k \cup \{\pi \in \mathcal{A}_k \backslash \mathcal{C}_k : \widehat{FDR}(\pi) + C_1(\pi, t, \delta_k)/|\pi| \leq \alpha\}$

    Set $\mathcal{A}_{k+1} = \mathcal{A}_k$

    Remove any $\pi$ from $\mathcal{A}_{k+1}$ and $\mathcal{C}_{k+1}$ such that one of the conditions is true:

     1. $\widehat{FDR}(\pi) - C_1(\pi, t, \delta_k)/|\pi| > \alpha$

     2. $\exists \pi' \in \mathcal{C}_{k+1}$ with $\widehat{TP}(\pi') - \widehat{TP}(\pi) > C_2(\pi, \pi', t, \delta_k)$ and add $\pi$ to a set $R$

    Remove any $\pi$ from $\mathcal{A}_{k+1}$ and $\mathcal{C}_{k+1}$ such that:

     3. $\exists \pi' \in \mathcal{C}_{k+1} \cup R$, such that $\pi \subset \pi'$.

    Set $S_{k+1} := \bigcup_{\pi \in \mathcal{A}_{k+1} \backslash \mathcal{C}_{k+1}} \pi$, and $T_{k+1} = \bigcup_{\pi \in \mathcal{A}_{k+1}} \pi - \bigcap_{\pi \in \mathcal{A}_{k+1}} \pi$.

    $k \leftarrow k + 1$

  **endif**

  **Stochastic Noise:**

    if $|\mathcal{A}_k| = 1$, **Break**. Otherwise:

    Sample $I_t \sim \text{Unif}([n])$. If $I_t \in S_k$, then receive a label $Y_{I_t, t} \sim \text{Ber}(\eta_{I_t})$.

    Sample $J_t \sim \text{Unif}([n])$. If $J_t \in T_k$, then receive a label $Y'_{J_t, t} \sim \text{Ber}(\eta_{J_t})$.

  **Persistent Noise:**

    If $|\mathcal{A}_k| = 1$ or $t > n$, **Break.** Otherwise:

    Sample $I_t \sim [n] \backslash \{I_s : 1 \leq s < t\}$. If $I_t \in S_k$, then receive a label $Y_{I_t, t} = \eta_{I_t}$.

    Sample $J_t \sim [n] \backslash \{J_s : 1 \leq s < t\}$. If $J_t \in T_k$, then receive a label $Y'_{J_t, t} = \eta_{J_t}$.

  Return $\max_{t \in \mathcal{C}_{k+1}} \widehat{TP}(\pi)$

</div>

Algorithm 2: Active FDR control in persistent and bounded noise settings.

between classification and FDR-control. In one we choose to sample to maximize $\eta_\pi$ subject to FDR constraints where each $\eta_i \in [0, 1]$, whereas in classification we are trying to maximize $\mu_\pi$ where each $\mu_i \in [-1, 1]$. A major consequence of this difference is that $\eta_\pi \leq \eta_{\pi'}$ whenever $\pi \subseteq \pi'$, but such a condition does not hold for $\mu_\pi, \mu_{\pi'}$.

**Motivating the sample complexity:** As mentioned above, the general combinatorial bandit problem is considered in [11]. There they present an algorithm with sample complexity,

$$C \sum_{i=1}^n \max_{\pi: i \in \pi \Delta \pi^*} \frac{1}{|\pi \Delta \pi^*|} \frac{1}{\widetilde{\Delta}_\pi^2} \log\left( \max(|B(|\pi \Delta \pi^*|, \pi)|, |B(|\pi \Delta \pi^*|, \pi^*)|) \frac{n}{\delta} \right)$$

This complexity parameter is difficult to interpret directly so we compare it to one more familiar in statistical learning - the VC dimension. To see how this sample complexity relates to ours in Theorem 1, note that $\log_2 |B(k, \pi^*)| \leq \log_2 \binom{n}{k} \lesssim k \log_2(n)$. Thus by the Sauer-Shelah lemma, $V(B(r, \pi^*)) \lesssim \log_2(|B(r, \pi^*)|) \lesssim \min\{V(B(r, \pi^*)), r\} \log_2(n)$ where $\lesssim$ hides a constant. The proof of the confidence intervals in the supplementary effectively combines these two facts along with a union bound over all sets in $B(r, \pi^*)$.

## 4 Combinatorial FDR Control

Algorithm 2 provides an active sampling method for determining $\pi \in \Pi$ with $FDR(\pi) \leq \alpha$ and maximal $TPR$, which we denote as $\pi_\alpha^*$. Since $TPR(\pi) = TP(\pi)/\eta_{[n]}$, we can ignore the denominator and so maximizing the $TPR$ is the same as maximizing $TP$. The algorithm proceeds in epochs. At all times a collection $\mathcal{A}_k \subseteq \Pi$ of active sets is maintained along with a collection of FDR-controlled sets $\mathcal{C}_k \subseteq \mathcal{A}_k$. In each time step, random indexes $I_t$ and $J_t$ are sampled from the union $S_k = \cup_{\pi \in \mathcal{A}_k \backslash \mathcal{C}_k} \pi$ and the symmetric difference $T_k = \cup_{\pi \in \mathcal{A}_k} \pi - \cap_{\pi \in \mathcal{A}_k} \pi$ respectively. Associated random labels $Y_{I_t, t}, Y_{J_t, t} \in \{0, 1\}$ are then obtained from the underlying label distributions $\text{Ber}(\eta_{I_t})$ and $\text{Ber}(\eta_{J_t})$. At the start of each epoch, any set with a $FDR$ that is statistically known to be

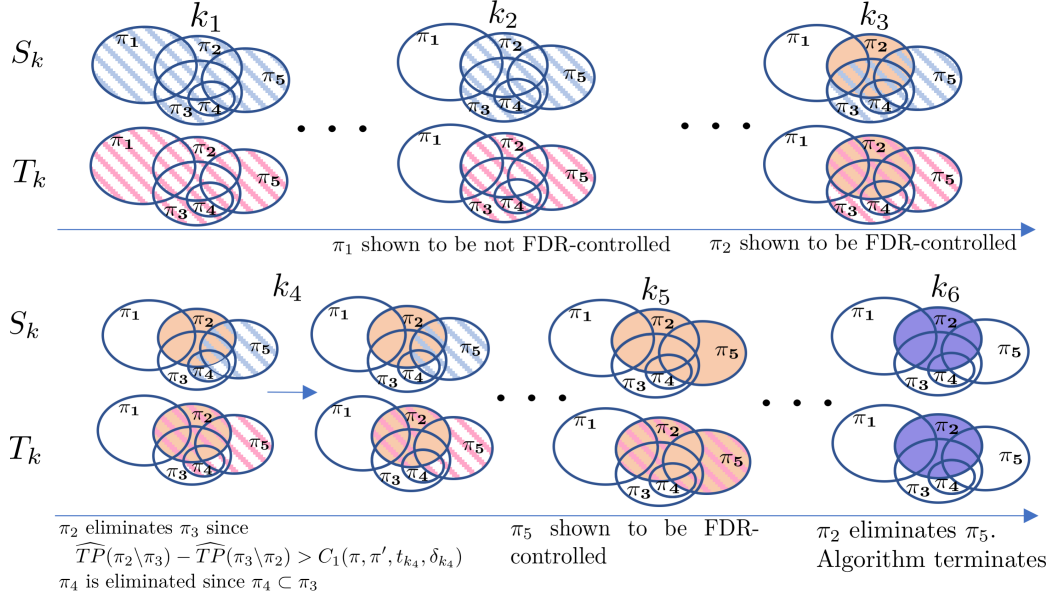

Figure 2: Example run of Algorithm 2, showing the evolution of sampling regions $S_k$ (blue stripes), $T_k$ (pink stripes) and FDR controlled sets $C_k$ (orange fill) at each time $k_t$.

under $\alpha$ is added to $\mathcal{C}_k$, and any sets whose $FDR$ are greater than $\alpha$ are removed from $\mathcal{A}_k$ in condition 1. Similar to the active classification algorithm of Figure 1, a set $\pi \in \mathcal{A}_k$ is removed in condition 2 if $TP(\pi)$ is shown to be statistically less than $TP(\pi')$ for some $\pi' \in \mathcal{C}_k$ that, crucially, is FDR controlled. In general there may be many sets $\pi \in \Pi$ such that $TP(\pi) > TP(\pi_\alpha^*)$ that are not FDR-controlled. Finally in condition 3, we exploit the positivity of the $\eta_i$'s: if $\pi \subset \pi'$ then deterministically $TP(\pi) \leq TP(\pi')$, so if $\pi'$ is FDR controlled it can be used to eliminate $\pi$. The choice of $T_k$ is motivated by active classification: we only need to sample in the symmetric difference. To determine which sets are FDR-controlled it is important that we sample in the entirety of the union of all $\pi \in \mathcal{A}_k \setminus \mathcal{C}_k$, not just the symmetric difference of the $\mathcal{A}_k$, which motivates the choice of $S_k$. In practical experiments persistent noise is not uncommon and avoids the potential for unbounded sample complexities that potentially occur when $FDR(\pi) \approx \alpha$. Figure 2 demonstrates a model run of the algorithm in the case of five sets $\Pi = \{\pi_1, \ldots, \pi_5\}$.

Recall that $\Pi_\alpha$ is the subset of $\Pi$ that is FDR-controlled so that $\pi_\alpha^* = \arg\max_{\pi \in \Pi_\alpha} TP(\pi)$. The following gives a sample complexity result for the number of rounds before the algorithm terminates.

**Theorem 2** *Assume that for each $i \leq n$ there is an associated $\eta_i \in [0,1]$ and $\{Y_{i,j}\}_{j=1}^\infty$ is an i.i.d. sequence of random variables such that $Y_{i,j} \sim Ber(\eta_i)$. For any $\pi \in \Pi$ define $\Delta_{\pi,\alpha} = |FDR(\pi) - \alpha|$, and $\widetilde{\Delta}_\pi = |TP(\pi_\alpha^*) - TP(\pi)|/|\pi \Delta \pi^*| = |TP(\pi_\alpha^* \setminus \pi) - TP(\pi \setminus \pi_\alpha^*)|/|\pi \Delta \pi^*|$, and*

$$s_\pi^{FDR} = \frac{V_\pi}{|\pi|} \frac{1}{\Delta_{\pi,\alpha}^2} \log\left(n \log(\Delta_{\pi,\alpha}^{-2})/\delta\right), \quad s_\pi^{TP} = \frac{V_{\pi,\pi_\alpha^*}}{|\pi \Delta \pi_\alpha^*|} \frac{1}{\widetilde{\Delta}_\pi^2} \log\left(n \log(\widetilde{\Delta}_\pi^{-2})/\delta\right)$$

*In addition define $T_\pi^{FDR} = \min\{s_\pi^{FDR}, \max\{s_\pi^{TP}, s_{\pi_\alpha^*}^{FDR}\}, \min_{\substack{\pi' \in \Pi_\alpha \\ \pi \subset \pi'}} s_{\pi'}^{FDR}\}$ and*

$$T_\pi^{TP} = \min\{\max\{s_\pi^{TP}, s_{\pi_\alpha^*}^{FDR}\}, \min_{\substack{\pi' \in \Pi_\alpha \\ \pi \subset \pi'}} s_{\pi'}^{FDR}\}. \; Using \; C_1(\pi, t, \delta) := \sqrt{\frac{4|\pi| n V_\pi \log\left(\frac{n}{\delta}\right)}{t}} +$$

$\frac{4 n V_\pi \log\left(\frac{n}{\delta}\right)}{3t}$ *and $C_2 = C$ for $C$ defined in Theorem 1, for a fixed constant $c$, with probability at least $1 - \delta$, in the **stochastic noise** setting Algorithm 2 returns $\pi_\alpha^*$ after a number of samples no more than*

$$c \underbrace{\sum_{i=1}^n \max_{\pi \in \Pi: i \in \pi} T_\pi^{FDR}}_{FDR-Control} + c \underbrace{\sum_{i=1}^n \max_{\pi \in \Pi_\alpha: i \in \pi \Delta \pi_\alpha^*} T_\pi^{TP}}_{TPR-Elimination}$$

*and in the **persistent noise** setting returns $\pi_\alpha^*$ after no more than* $c \sum_{i=1}^{n} \min \left\{ 1, \left( \max_{\pi \in \Pi : i \in \pi} T_\pi^{FDR} + \max_{\pi \in \Pi_\alpha : i \in \pi \Delta \pi_\alpha^*} T_\pi^{TP} \right) \right\}$

Though this result is complicated, each term is understood by considering each way a set can be removed and the time at which an arm $i$ will stop being sampled. Effectively the sample complexity decomposes into two parts, the complexity of showing that a set is FDR-controlled or not, and how long it takes to eliminate it based on TPR. To motivate $s_\pi^{FDR}$, if we have a single set $\pi$ then $1/(|\pi|\Delta_{\pi,\alpha}^2)$ roughly captures the number of times we have to sample each element in $\pi$ to decide whether it is FDR-controlled or not - so in particular in the general case we have to roughly sample an arm $i$, $\max_{\pi \in \Pi, i \in \pi} s_\pi$ times. However, we can remove a set before showing it is FDR controlled using other conditions which $T_\pi^{FDR}$ captures. The term in the sample complexity for elimination using TPR is similarly motivated. We now unpack the underbraced terms more carefully simultaneously explaining the sample complexity and the motivation for the proof of Theorem 2.

**Sample Complexity of FDR-Control** In any round where there exists a set $\pi \in \mathcal{A}_k \setminus \mathcal{C}_k$ with arm $i \in \pi$, i.e. $\pi$ is not yet FDR controlled, there is the potential for sampling $i \in \mathcal{S}_k$. A set $\pi$ only leaves $\mathcal{A}_k$ if $i$) it is shown to not be FDR controlled (condition 1 of the algorithm), $ii$) because an FDR controlled set eliminates it on the basis of TP (condition 2), or $iii$) it is contained in an FDR controlled set (condition 3). These three cases reflect the three arguments of the $\min$ in the defined quantity $T_\pi^{FDR}$, respectively. Taking the maximum over all sets containing an arm $i$ and summing over all $i$ gives the total FDR-control term. This is a large savings relative to naive non-adaptive algorithms that sample until every set $\pi$ in $\Pi$ was FDR controlled which would take $O(n \max_{\pi \in \Pi} s_\pi^{FDR})$ samples.

**Sample Complexity of TPR-Elimination** An FDR-controlled set $\pi \in \Pi_\alpha$ is only removed from $\mathcal{C}_k$ when eliminated by an FDR-controlled set with higher $TP$ or if it is removed because it is contained in an FDR-controlled set. In general we can upper bound the former time by the samples needed for $\pi_\alpha^*$ to eliminate $\pi$ once we know $\pi_\alpha^*$ is FDR controlled - this gives rise to $\max_{\pi \in \Pi_\alpha : i \in \pi \Delta \pi_\alpha^*} T_\pi^{TP}$. Note that sets are removed in a procedure mimicking active classification and so the active gains there apply to this setting as well. A naive passive algorithm that continues to sample until both the FDR of every set is determined, and $\pi_\alpha^*$ has higher TP than every other FDR-controlled set gives a significantly worse sample complexity of $O(n \max\{\max_{\pi \in \Pi_\alpha} s_\pi^{FDR}, \max_{\pi \notin \Pi_\alpha} s_\pi^{TP}\})$.

**Comparison with [5].** Similar to our proposed algorithm, [5] samples in the union of all active sets and maintains statistics on the empirical FDR of each set, along the way removing sets that are not FDR-controlled or have lower TPR than an FDR-controlled set. However, they fail to sample in the symmetric difference, missing an important link between FDR-control and active classification. In particular, the confidence intervals they use are far looser as a result. They also only consider the case of persistent noise. Their proven sample complexity results are no better than those achieved by the passive algorithm that samples each item uniformly, which is precisely the sample complexity described at the end of the previous paragraph.

**One Dimensional Thresholds** Consider a stylized modeling of the topology $\beta\alpha\beta\beta$ from the introduction in the persistent noise setting where $\Pi = \{[t] : t \leq n\}$, $\eta_i \sim \text{Ber}(\beta \mathbf{1}\{i \leq z\})$ with $\beta < .5$, and $z \in [n]$ is assumed to be small, i.e., we assume that there is only a small region in which positive labels can be found and the Bayes classifier is just to predict 0 for all points. Assuming $\alpha > 1 - \beta$, one can show the sample complexity of Algorithm 2 satisfies $O((1-\alpha)^{-2}(\log(n/(1-\alpha)) + (1+\beta)z/(1-\alpha)))$ while any naive non-adaptive sampling strategy will take at least $O(n)$ samples.

**Implementation.** For simple classes $\Pi$ such as thresholds or axis aligned rectangles, our algorithm can be made computationally efficient. But for more complex classes there may be a wide gap between theory and practice, just as in classification [36, 20]. However, the algorithm motivates two key ideas - sample in the union of potentially good sets to learn which are FDR controlled, and sample in the symmetric difference to eliminate sets. The latter insight was originally made by $A^2$ in the case of classification and has justified heuristics such as uncertainty sampling [36]. Developing analogous heuristics for the former case of FDR-control is an exciting avenue of future work.

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
