[Supplementary Material · supp.pdf]

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

171 our sampling strategy is uniformly drawing from $[n]$ in each round, but only paying to see a label
172 if $I_t \in T_k$, the underlying sampling distribution is still uniform regardless of the round and so the
173 estimate of $\widehat{\mu}_{\pi',k} - \widehat{\mu}_{\pi,k}$ is unbiased. In practice, since the number of samples that land in $T_k$ follow

174  a geometric distribution, instead of using rejection sampling we could instead have drawn a single
175  sample from a geometric distribution and sampled that many uniformly at random from $T_k$.

---

**Input**: $\delta$, $\Pi \subset 2^{[n]}$, Confidence bound $C(\pi', \pi, t, \delta)$.
Let $\mathcal{A}_1 = \Pi$, $T_1 = (\cup_{\pi \in \mathcal{A}_1} \pi) - (\cap_{\pi \in \mathcal{A}_1} \pi)$, $k = 1$, $\mathcal{A}_k$ will be the active sets in round $k$
**for** $t = 1, 2, \cdots$
   **if** $t == 2^k$**:**
     Set $\delta_k = .5\delta/k^2$. For each $\pi, \pi'$ let
     $\widehat{\mu}_{\pi',k} - \widehat{\mu}_{\pi,k} = \frac{n}{t}(\sum_{s=1}^t R_{I_s,s} \mathbf{1}\{I_s \in \pi' \setminus \pi\} - \sum_{s=1}^t R_{I_s,s} \mathbf{1}\{I_s \in \pi \setminus \pi'\})$
     Set $\mathcal{A}_{k+1} = \mathcal{A}_k - \{\pi \in \mathcal{A}_k : \exists \pi' \in \mathcal{A}_k \text{with } \widehat{\mu}_{\pi',k} - \widehat{\mu}_{\pi,k} > C(\pi', \pi, t, \delta_k)\}$.
     Set $T_{k+1} = (\cup_{\pi \in \mathcal{A}_{k+1}} \pi) - (\cap_{\pi \in \mathcal{A}_{k+1}} \pi)$.
     $k \leftarrow k + 1$
   **endif**
   **Stochastic Noise:**
     If $T_k = \emptyset$, **Break**. Otherwise, draw $I_t$ uniformly at random from $[n]$ and if $I_t \in T_k$ receive an
     associated reward $R_{I_t,t} = 2Y_{I_t,t} - 1, Y_{I_t,t} \overset{iid}{\sim} \text{Ber}(\eta_{I_t})$.
   **Persistent Noise:**
     If $T_k = \emptyset$ or $t > n$, **Break**. Otherwise, draw $I_t$ uniformly at random from $[n] \setminus \{I_s : 1 \leq s < t\}$
     and if $I_t \in T_k$ receive associated reward $R_{I_t,t} = 2Y_{I_t,t} - 1, Y_{I_t,t} = \eta_{I_t}$.
**Output:** $\pi' \in \mathcal{A}_k$ such that $\widehat{\mu}_{\pi',k} - \widehat{\mu}_{\pi,k} \geq 0$ for all $\pi \in \mathcal{A}_k \setminus \pi'$

Algorithm 1: Action Elimination for Active Classification

176  For any $\mathcal{A} \subseteq 2^{[n]}$ define $V(\mathcal{A})$ as the VC-dimension of a collection of sets $\mathcal{A}$. Given a family of sets,
177  $\Pi \subseteq 2^{[n]}$, define $B_1(k) := \{\pi \in \Pi : |\pi| = k\}$, $B_2(k, \pi') := \{\pi \in \Pi : |\pi \Delta \pi'| = k\}$. Also define
178  the following complexity measures:

$$V_\pi := V(B_1(|\pi|)) \wedge |\pi| \text{ and } V_{\pi,\pi'} := \max\{V(B_2(|\pi \Delta \pi'|, \pi)), V(B_2(|\pi \Delta \pi'|, \pi'))\} \wedge |\pi \Delta \pi'|$$

179  In general $V_\pi, V_{\pi,\pi'} \leq V(\Pi)$. A contribution of our work is the development of confidence intervals
180  that do not depend on a union bound over the class but instead on local VC dimensions. These are
181  described carefully in Lemma 1 in the supplementary materials.

182  **Theorem 1** *For each $i \in [n]$ let $\mu_i \in [-1, 1]$ be fixed but unknown and assume $\{R_{i,j}\}_{j=1}^\infty$ is an*
183  *i.i.d sequence of random variables such that $\mathbb{E}[R_{i,j}] = \mu_i$ and $R_{i,j} \in [-1, 1]$. Define $\widetilde{\Delta}_\pi =$*
184  *$|\mu_\pi - \mu_{\pi^*}|/|\pi \Delta \pi^*|$, and*

$$\tau_\pi = \frac{V_{\pi,\pi^*}}{|\pi^* \Delta \pi|} \frac{1}{\widetilde{\Delta}_\pi^2} \log\left(n \log(\widetilde{\Delta}_\pi^{-2})/\delta\right).$$

185  *Using $C(\pi, \pi', t, \delta) := \sqrt{\frac{8|\pi \Delta \pi'| n V_{\pi,\pi'} \log\left(\frac{n}{\delta}\right)}{t}} + \frac{4 n V_{\pi,\pi'} \log\left(\frac{n}{\delta}\right)}{3t}$ for a fixed constant c, with probability*
186  *greater than $1 - \delta$, in the **stochastic noise** setting Algorithm 1 returns $\pi_*$ after a number of samples*
187  *no more than $c \sum_{i=1}^n \max_{\pi \in \Pi : i \in \pi \Delta \pi^*} \tau_\pi$ and in the **persistent noise** setting the number of samples*
188  *needed is no more than $c \sum_{i=1}^n \min\{1, \max_{\pi \in \Pi : i \in \pi \Delta \pi^*} \tau_\pi\}$*

189  One always has $1/|\pi^* \Delta \pi| \leq V_{\pi,\pi^*}/|\pi^* \Delta \pi| \leq 1$ and both bounds are achievable by different classes
190  $\Pi$. In addition, in terms of risk $\widetilde{\Delta}_\pi = |\mu_\pi - \mu_{\pi^*}|/|\pi \Delta \pi^*| = n|R(\pi) - R(\pi^*)|/|\pi \Delta \pi^*|$. Since
191  sampling is done without replacement for persistent noise, there are improved confidence intervals
192  that one can use in that setting described in Lemma 1 in the supplementary materials. Finally, if we
193  had sampled non-adaptively, i.e. without rejection sampling, we would have had a sample complexity
194  of $O(n \max_{i \in [n]} \max_{\pi : \Pi : i \in \pi \Delta \pi^*} \tau_\pi)$.

195  **Remark:** Our rewards could be drawn from arbitrary distributions, not just Bernoulli label distri-
196  butions. In fact if we allow $R_{I_t,t} \sim^{i.i.d.} \nu_i$, where $\nu_i$ is a distribution supported on $[-1, 1]$ with
197  $\mathbb{E}[\nu_i] = \mu_i$, then Algorithm 1 gives state of the art results for the more general *pure exploration*
198  *combinatorial bandit* problem and furthermore Theorem 1 holds verbatim. Algorithm 1 is similar
199  to previous action elimination algorithms for combinatorial bandits in the literature, e.g. Algorithm
200  4 in [10]. However, unlike previous algorithms, we do not insist on sampling each item once, an
201  unrealistic requirement for classification settings. We discuss this connection further in Section A.1
202  in the supplementary materials and prove Theorem 1 in this more general setting.

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

It's natural to ask whether the $\log(n)$ on the right can be dropped. In specific examples, like nested classes, tools from empirical process theory (see Theorem 13.7 in [6]) imply that it can be improved to a $\log \log(n)$. We give such an example where the $\log(n)$ is not necessary in Appendix C for the case of one-dimensional thresholds.

## A.2 Confidence Bounds for Combinatorial Bandits

In this section, we build confidence intervals useful in our general combinatorial bandit setup discussed in the previous section. The union bounds presented are motivated by those in [10]. The constants used in the case without replacement are motivated by Corollary 3.6 in [3].

**Lemma 1** *Assume that for each arm $i \leq n$ there is an associated distribution $\nu_i$ with support $[-1, 1]$, mean $\mu_i$ and variance $\sigma_i^2 \leq 1$. Assume access to the observations $(I_1, y_{I_1}) \cdots, (I_t, y_{I_t})$ in two different but related settings, let $s \leq t$,*

444     1. **Stochastic Noise** $I_s \sim Unif([n])$ and $y_{I_s} \sim \nu_{I_s}$.

445     2. **Persistent Noise** $I_s \in [n]$ are drawn without replacement, $y_{I_s} = \mu_{I_s}$, $s \leq n$

446     *Let* $\widehat{\mu}_\pi = \frac{n}{T}\sum_{k=1}^{T} y_s \mathbf{1}\{I_s \in \pi\}$. *Then*

447         *1. With probability greater than* $1 - \delta$ *for all* $\pi \in \Pi$

$$|\widehat{\mu}_\pi - \mu_\pi| \leq C_1(\pi, t, \delta) := \sqrt{\frac{4\rho_t |\pi| n V_\pi \log\left(\frac{n}{\delta}\right)}{t}} + \frac{4n\kappa_t V_\pi \log\left(\frac{n}{\delta}\right)}{3t} \tag{1}$$

448         *2. Fix* $\pi' \in \Pi$. *With probability greater than* $1 - \delta$ *for all* $t > 0$ *and* $\pi \in \Pi$

$$\left|\widehat{\mu}_{\pi'\backslash\pi} - \widehat{\mu}_{\pi\backslash\pi'} - (\mu_{\pi'\backslash\pi} - \mu_{\pi\backslash\pi'})\right| \leq C_2(\pi, \pi', t, \delta) := \sqrt{\frac{8\rho_t |\pi\Delta\pi'| n V_{\pi,\pi'} \log\left(\frac{n}{\delta}\right)}{t}} \tag{2}$$

$$+ \frac{4\kappa_t n V_{\pi,\pi'} \log\left(\frac{n}{\delta}\right)}{3t} \tag{3}$$

449     *where* $\rho_t, \kappa_t = 1$ *in the stochastic case and in the persistent case*

$$\rho_t = \begin{cases} 1 - \frac{t-1}{n} & t \leq n/2 \\ 1 - \frac{t}{n} & t \geq n/2 \end{cases} \qquad \kappa_t = \frac{4}{3} + \begin{cases} \sqrt{\frac{t(t-1)}{n(n-t+1)}} & t \leq n/2 \\ \sqrt{\frac{(n-t-1)(n-t)}{(t+1)n}} & t \geq n/2 \end{cases}$$

450   Note that by negative associativity the confidence bounds that hold in the case of sampling with
451   replacement also hold when sampling without replacement.

452   **Proof:**    Define the complexity measures

$$B_1(k) = \{\pi \in \mathcal{A} : |\pi| = k\}, B_2(k, \pi') = \{\pi \in \mathcal{A} : |\pi\Delta\pi'| = k\}.$$

453   Firstly note that for any $\pi \in \Pi$

$$\begin{aligned}
\mathrm{var}(\widehat{\mu}_\pi) &= \frac{n^2}{T} \mathrm{var}\left(y_1 \mathbf{1}\{I_1 \in \pi\}\right) \\
&= \frac{n^2}{T}\left(\mathbb{E}[y_1^2 \mathbf{1}\{I_1 \in \pi \backslash \pi'\}] - \left(\frac{1}{n}\sum_{i \in \pi\backslash\pi'} \mu_i\right)^2\right) \\
&\leq \frac{n^2}{T}\left(\frac{1}{n}\sum_{i \in \pi}(\sigma_i^2 + \mu_i^2)\right) \leq \frac{2|\pi|n}{T}
\end{aligned}$$

454   Thus by Bernstein's inequality and a union bound,

$$\mathbb{P}\left(\exists \pi \in \Pi : |\widehat{\mu}_\pi - \mu_\pi| > \sqrt{\frac{2|\pi|n \log(nB_1(|\pi|)/\delta)}{T}} + \frac{2n \log(nB_1(|\pi|)/\delta)}{3T}\right) \leq \sum_{\pi \in \Pi} \frac{\delta}{nB_1(|\pi|)}$$

$$\leq \sum_{k=1}^{n} B_1(k) \frac{\delta}{nB_1(k)} \leq \delta$$

455 For the second assertion, firstly note that for any $\pi, \pi'$, $\widehat{\mu}_\pi - \widehat{\mu}_{\pi'} = \widehat{\mu}_{\pi \setminus \pi'} - \widehat{\mu}_{\pi' \setminus \pi}$ and so

$$
\begin{aligned}
&\mathrm{var}(\widehat{\mu}_\pi - \widehat{\mu}_{\pi'}) \\
&= \mathrm{var}(\widehat{\mu}_{\pi \setminus \pi'} - \widehat{\mu}_{\pi' \setminus \pi}) \\
&= \mathrm{var}(\widehat{\mu}_{\pi \setminus \pi'}) + \mathrm{var}(\widehat{\mu}_{\pi' \setminus \pi})) \\
&= \frac{n^2}{T} \mathrm{var}\left(y_1 \mathbf{1}\{I_1 \in \pi \setminus \pi'\}\right) + \frac{n^2}{T} \mathrm{var}\left(y_1 \mathbf{1}\{I_1 \in \pi \setminus \pi'\}\right) \\
&= \frac{n^2}{T}\left(\mathbb{E}[y_1^2 \mathbf{1}\{I_1 \in \pi \setminus \pi'\}] - \left(\frac{1}{n}\sum_{i \in \pi \setminus \pi'} \mu_i\right)^2 + \mathbb{E}[y_1^2 \mathbf{1}\{I_1 \in \pi' \setminus \pi\}] - \left(\frac{1}{n}\sum_{i \in \pi' \setminus \pi} \mu_i\right)^2\right) \\
&\le \frac{n^2}{T}\left(\frac{1}{n}\sum_{i \in \pi \setminus \pi'}(\sigma_i^2 + \mu_i^2) + \frac{1}{n}\sum_{i \in \pi' \setminus \pi}(\sigma_i^2 + \mu_i^2)\right) \\
&\le \frac{4|\pi \Delta \pi'|n}{T}
\end{aligned}
$$

456 Let $b_\pi = \max\{|B_2(|\pi \Delta \pi'|, \pi)|, |B_2(|\pi \Delta \pi'|, \pi')|)\}$

$$
\begin{aligned}
&\mathbb{P}\left(\exists \pi \in \Pi : |\widehat{\mu}_{\pi' \setminus \pi} - \widehat{\mu}_{\pi \setminus \pi'} - \mu_{\pi' \setminus \pi} - \mu_{\pi \setminus \pi'}| > \sqrt{\frac{8|\pi \Delta \pi'|\log(n b_\pi/\delta)}{T}} + \frac{2n \log(b_\pi/\delta)}{3T}\right) \\
&\qquad\le \sum_{\pi \in \Pi} \frac{\delta}{n b_\pi} \\
&\qquad\le \sum_{k=1}^n \sum_{\pi \in \Pi} \mathbf{1}\{|\pi \Delta \pi'| = k\} \frac{\delta}{n b_\pi} \\
&\qquad= \sum_{k=1}^n \sum_{\pi \in \Pi} \mathbf{1}\{|\pi \Delta \pi'| = k\} \frac{\delta}{n \max\{|B_2(|\pi \Delta \pi'|, \pi)|, |B_2(|\pi \Delta \pi'|, \pi')|\}} \\
&\qquad\le \sum_{k=1}^n \sum_{\pi \in \Pi} \mathbf{1}\{|\pi \Delta \pi'| = k\} \frac{\delta}{n |B_2(|\pi \Delta \pi'|, \pi')|} \\
&\qquad\le \sum_{k=1}^n \frac{\delta}{n} \le \delta
\end{aligned}
$$

457 Now by the Sauer-Shelah Lemma for any $k$

$$
\log(B_1(k)) \le V(B_1(k)) \log(en/V(B_1(k))).
$$

458 where $V(\cdot)$ denotes the VC-dimension. At the same time, $|B_1(k)| \le |\{\pi \in \Pi : |\pi| = k\}| \le n^k$.
459 Hence

$$
\begin{aligned}
\log(n|B_1(k)|/\delta) &\le \min\{V(B_1(k)) \log(en/V(B_1(k))) + \log(n/\delta), (k+1)\log(n/\delta)\} \\
&\le 4 \min\{V(B_1(k)), k\} \log(en/\delta)
\end{aligned}
$$

460 Similarly for any $k$,

$$
\log(B_2(k, \pi')) \le V(B_2(k, \pi')) \log(en/V(B_2(k, \pi')))
$$

461 and $|\{\pi \in \Pi : |\pi \Delta \pi^*| = k\}| = \binom{n}{k} \le n^k$. In particular,

$$
\begin{aligned}
\log(n|B_2(k, \pi')|/\delta) &\le \min\{V(B_2(k, \pi')) \log(en/V(B_2(k, \pi'))) + \log(n/\delta), (k+1)\log(n/\delta)\} \\
&\le 4 \min\{V(B_2(k, \pi')), k\} \log(en/\delta)
\end{aligned}
$$

462 So using identical logic

$$
\begin{aligned}
\log(n b_\pi/\delta) &\le \log(n \max\{|B_2(|\pi \Delta \pi'|, \pi)|, |B_2(|\pi \Delta \pi'|, \pi')|)\}/\delta) \\
&\le \max\{\log(n|B_2(|\pi \Delta \pi'|, \pi)|/\delta), \log(n|B_2(|\pi \Delta \pi'|, \pi')|/\delta)\} \\
&\le 4 \min\{\max\{V(B_2(|\pi \Delta \pi'|, \pi)), V(B_2(|\pi \Delta \pi'|, \pi'))\}, |\pi \Delta \pi'|\} \log(en/\delta)
\end{aligned}
$$

463 Finally, in the case of without replacement, we can use the confidence intervals from Theorem 3.6 of
464 [3] and the result follows. □

## A.3 Comparison to the Disagreement Coefficient

One of the foundational works on active learning is the DHM algorithm of [19] and the $A^2$ algorithm that preceded it [2]. In their setting a set of points, $x_1, x_2, \cdots$ are streamed to a learner who chooses whether to label a point or not. Similar in spirit to our algorithm, DHM determines whether it is certain or not about how $\pi^*$ would label the current point, and if not, would request the label. Thus, DHM only requests the labels of any point that it is uncertain about given all the information up to that time. A key quantity arising in the sample complexity of DHM (and many previous works on active classification) has been that of the disagreement coefficient of the set $\pi^*$: $\theta = \theta(\epsilon, \pi^*) :=$ $\sup_{r \geq n(\epsilon+\nu)} \left\{ \frac{|x:x \in \pi \Delta \pi^*, \pi \in \Pi \text{ and } |\pi \Delta \pi^*| \leq r|}{r} \right\}$ where $\nu = \mathbb{P}(\pi^*(x) \neq y)$ and $\epsilon$ is a bound on the excess error of the set $\widehat{\pi}$ returned by an active learning algorithm. After being streamed $m$ points, DHM returns a classifier with error at most $O(\nu + V(\Pi) \log(m/\delta)/m + \sqrt{V(\Pi)\nu \log(m/\delta)/m})$ after labeling $O\left( \theta \left( \nu m + V(\Pi) \log^2(m) + \log\left( \frac{\log(m)}{\delta} \right) \right) \right)$ samples (provided $\epsilon \leq \nu$–the realistic setting in the non-realizable noisy case). Ignoring log factors, this roughly says that a classifier with error at most $\nu + \epsilon$ is returned after $\theta V(\Pi) \nu \max\{\epsilon^{-1}, \nu\epsilon^{-2}\}$ requested labels.

In general the analysis of the DHM algorithm can not characterize the contribution of each arm to the overall sample complexity leading to sub-optimal sample complexity for combinatorial classes. Consider the case when $\Pi = \{\pi_i\}_{i=1}^n$, with $\pi_i = \{i\}$, and $\pi^* = \{i^*\}$ where $i^* = \text{argmax}_{i \leq n} \mu_i$. If we take $\mu_i \in [-1/2, 1/2]$ for all $i$ then $\frac{1}{4} - \frac{1}{2n} \leq \nu \leq \frac{3}{4} + \frac{1}{2n}$ and for best-arm we necessarily have $\epsilon = \min_{j \neq i^*} \frac{1}{n} (\mu_{i^*} - \mu_j)$. One can show for this problem $\theta = \frac{1}{\nu+\epsilon}$ and so the bound of Theorem 1 of [19] scales like $\theta d\nu \max\{\epsilon^{-1}, \nu\epsilon^{-2}\} = \frac{\nu}{\nu+\epsilon} \max\{\epsilon^{-1}, \nu\epsilon^{-2}\} \approx \epsilon^{-2} = n^2 \max_{i \neq i_*} \Delta_i^{-2}$ for $\Delta_i = \mu_{i^*} - \mu_i$, which is substantially worse than our bound for this problem which scales like $\sum_{i \neq i_*} \Delta_i^{-2}$, describing the contribution from each individual item. Similar arguments can be made for other combinatorial classes such as all subsets of size $k$. We emphasize that it is not that we are particularly interested in applying algorithms like DHM to this specific problem, but that it exposes such a gross inconsistency with the best known algorithms that its application in general should be questioned.

## A.4 Proof of Theorem 1

Since Active Classification is a specific case of the more general combinatorial bandit problem as described in A.1, we focus on the more general case throughout the following. Algorithm 1 is repeated in this more general case below - all that changes are the reward distributions are more general than just Bernoulli distributions.

**Proof:** Throughout the following, let $\Delta_\pi := \mu_{\pi^* \setminus \pi} - \mu_{\pi \setminus \pi^*}$. Define

$$\mathcal{E} = \bigcap_{k \in \mathbb{N}} \bigcap_{\pi \in \Pi} \{|(\widehat{\mu}_{\pi_*,k} - \widehat{\mu}_{\pi,k}) - (\mu_{\pi_*} - \mu_\pi)| \leq C(\pi_*, \pi, t_k, \delta_k)\}$$

where we recall $C(\pi_*, \pi, t_k, \delta_k) = C(\pi, \pi_*, t_k, \delta_k)$. By Lemma 1 we have that $\mathbb{P}(\mathcal{E}) \geq 1 - \sum_{k=1}^\infty \delta_k \geq 1 - \delta$ so assume $\mathcal{E}$ holds in what follows.

First we show $\pi_* \in \mathcal{A}_k$ for all $k$. Assume $\pi_* \in \mathcal{A}_k$. Then for any $\widehat{\pi} \in \mathcal{A}_k$ we have

$$\widehat{\mu}_{\widehat{\pi} \setminus \pi_*,k} - \widehat{\mu}_{\pi_* \setminus \widehat{\pi},k} \overset{\mathcal{E}}{\leq} \mu_{\widehat{\pi} \setminus \pi_*} - \mu_{\pi_* \setminus \widehat{\pi}} + C(\widehat{\pi}, \pi_*, t_k, \delta_k)$$
$$\leq C(\widehat{\pi}, \pi_*, t_k, \delta_k)$$

which implies that $\pi_* \in \mathcal{A}_{k+1}$. The result follows by the fact that $\pi_* \in \mathcal{A}_0$.

Now we bound the number of samples taken with high probability. For an arm $i$ to be sampled at time $t$, there must be at least two policies $\pi, \pi' \in \mathcal{A}_t$ such that $i \in \pi \Delta \pi'$. Since we just showed that $\pi_* \in \mathcal{A}_t$ for all $t$, it follows that $\min\left\{ k : \widehat{\mu}_{\pi_* \setminus \pi,k} - \widehat{\mu}_{\pi \setminus \pi_*,k} > C(\pi_*, \pi, t_k, \delta_k) \right\}$ is an upper bound on the number of rounds before $\pi$ is removed from $\Pi_t$. Since $\mu_{\pi_*} > \mu_\pi$ for all $\pi \in \Pi$, for each $\pi \in \Pi$ there exists a random first round $K_\pi$ when

$$\widehat{\mu}_{\pi_* \setminus \pi, K_\pi} - \widehat{\mu}_{\pi \setminus \pi_*, K_\pi} \geq C(\pi_*, \pi, t_{K_\pi}, \delta_{K_\pi}).$$

Algorithm 3: Action Elimination for Combinatorial Bandits

But for every $\pi \in \Pi$ and $k \in \mathbb{N}$ we have

$$\widehat{\mu}_{\pi_* \setminus \pi, k} - \widehat{\mu}_{\pi \setminus \pi_*, k} \overset{\mathcal{E}}{\geq} \Delta_\pi - C(\pi_*, \pi, t_k, \delta_k)$$

so define

$$k_\pi := \min\{k : \Delta_\pi/2 \geq C(\pi_*, \pi, t_k, \delta_k)\}.$$

Also define $k_{\max} = \max_\pi k_\pi$ and note that $k_{\max}$ is finite and deterministic since $C(\pi_*, \pi, t_k, \delta_k)$ is decreasing in $k$. Now we have that

$$S_k = \{i \in [n] : \exists \pi \in \Pi : i \in \pi_* \Delta \pi, K_\pi \geq k\}$$
$$\overset{\mathcal{E}}{\subseteq} \{i \in [n] : \exists \pi \in \Pi : i \in \pi_* \Delta \pi, k_\pi \geq k\}$$
$$=: s_k$$

Thus, we trivially have $\mathbf{1}\{I_s \in S_k\} \leq \mathbf{1}\{I_s \in s_k\}$ where the right hand side is a deterministic function. Furthermore, whether or not $I_s$ are drawn uniformly at random from $[n]$ (with replacement) or uniformly at random from $[n] \setminus \{i : I_s = i, 1 \leq s < t\}$ (without replacement for persistent noise), the $I_s$ indices are negatively associated random variables [20]. Consequently, standard multiplicative Chernoff bounds apply:

$$\mathbb{P}\left(\sum_{k=1}^{k_{\max}} \sum_{s=t_{k-1}+1}^{t_k} \mathbf{1}\{I_s \in S_k\} \geq (1+r) \sum_{k=1}^{k_{\max}} t_k \frac{|s_k|}{n}\right)$$

$$\leq \mathbb{P}\left(\sum_{k=1}^{k_{\max}} \sum_{s=t_{k-1}+1}^{t_k} \mathbf{1}\{I_s \in s_k\} \geq (1+r) \sum_{k=1}^{k_{\max}} t_k \frac{|s_k|}{n}\right)$$

$$\leq \exp\left(-\frac{\min\{r, r^2\}}{3} \sum_{k=1}^{k_{\max}} t_k \frac{|s_k|}{n}\right)$$

516　Taking $r = \max\left\{\frac{3\log(1/\delta)}{\sum_{k=1}^{k_{\max}} t_k \frac{|s_k|}{n}}, \sqrt{\frac{3\log(1/\delta)}{\sum_{k=1}^{k_{\max}} t_k \frac{|s_k|}{n}}}\right\}$ we have with probability at least $1 - \delta$ that

$$\sum_{k=1}^{k_{\max}} \sum_{s=t_{k-1}+1}^{t_k} \mathbf{1}\{I_s \in S_k\} \leq \max\left\{3\log(1/\delta), \sqrt{3\log(1/\delta)\sum_{k=1}^{k_{\max}} t_k \frac{|s_k|}{n}}\right\} + \sum_{k=1}^{k_{\max}} t_k \frac{|s_k|}{n}$$

$$\leq \frac{9}{2}\log(1/\delta) + \frac{3}{2}\sum_{k=1}^{\infty} t_k \frac{|s_k|}{n}$$

517　where the last inequality follows by the arithmetic-geometric mean inequality. Now

$$\sum_{k=1}^{\infty} t_k \frac{|s_k|}{n} = \sum_{k=1}^{\infty} t_k \sum_{i=1}^{n} \frac{1}{n}\mathbf{1}\{\exists \pi \in \Pi : i \in \pi_* \Delta\pi, k_\pi \geq k\}$$

$$= \sum_{k=1}^{\infty} \sum_{i=1}^{n} \frac{t_k}{n}\mathbf{1}\{\exists \pi \in \Pi : i \in \pi_* \Delta\pi, k_\pi \geq k\}$$

$$= \sum_{i=1}^{n} \sum_{k=1}^{\infty} \frac{2^k}{n}\mathbf{1}\{\exists \pi \in \Pi : i \in \pi_* \Delta\pi, 2^{k_\pi} \geq 2^k\}$$

$$\leq \sum_{i=1}^{n} \max_{\pi \in \Pi: i \in \pi_* \Delta\pi} \frac{2^{k_\pi+1}}{n}$$

518　Now, using the specific confidence interval $C_2(\pi', \pi, t_k, \delta_k)$ from 1

$$2^{k^\pi} \leq 2\min\{t \in \mathbb{N} : \Delta_\pi/2 < C_2(\pi_*, \pi, t, \delta_{\lceil \log_2 t \rceil})\}$$

$$\leq c_1 n V_{\pi,\pi'}\left(\frac{|\pi^*\Delta\pi|}{\Delta_\pi^2} + \frac{1}{\Delta_\pi}\right)\log\left(\frac{n\log\left(\Delta_\pi^{-2}\right)}{\delta}\right)$$

$$\leq c_2 n V_{\pi,\pi'}\frac{|\pi^*\Delta\pi|}{\Delta_\pi^2}\log\left(\frac{n\log\left(\Delta_\pi^{-2}\right)}{\delta}\right)$$

$$\leq c_2 \frac{n V_{\pi,\pi'}}{|\pi^*\Delta\pi|}\frac{1}{\widetilde{\Delta}_\pi^2}\log\left(\frac{n\log\left(\widetilde{\Delta}_\pi^{-2}\right)}{\delta}\right)$$

519　where the second to last line follows from

$$\frac{|\pi^*\Delta\pi|}{\Delta_\pi^2} + \frac{1}{\Delta_\pi} \leq \frac{1}{\Delta_\pi}\left(\frac{|\pi^*\Delta\pi|}{\Delta_\pi} + 1\right) \leq \frac{2|\pi^*\Delta\pi|}{\Delta_\pi^2}$$

520　since $\Delta_\pi \leq |\pi_*\Delta\pi|$. But for the persistent noise case we have $k_\pi \leq \log_2(n)$ which implies for any $i$,
521　$\max_{\pi \in \Pi: i \in \pi_* \Delta\pi} \frac{2^{k_\pi+1}}{n} \leq 2$. The result now follows.　　　□

## 522　B　Proof of Theorem 2

523　**Proof:**　**Step 1: Correctness** Let $t_k = 2^k$. Let $\mathcal{E}$ be the event that, for each $k$ and for each $\pi \in \Pi$,

$$\left|\widehat{FDR}(\pi) - FDR(\pi)\right| < C_1(\pi_t, n, t_k, \delta_k)/|\pi|$$

524　and

$$|(\widehat{TP}(\pi^* \setminus \pi) - \widehat{TP}(\pi \setminus \pi^*)) - (TP(\pi^* \setminus \pi) - TP(\pi \setminus \pi^*))| \leq C_2(\pi^*, \pi, t_k, \delta_k).$$

525　By Lemma 1 and a union bound,

$$\mathbb{P}(\mathcal{E}^c) \leq \sum_{k \geq 1} 2\frac{2\delta}{8k^2} \leq \delta$$

First we argue that $\pi^*$ is never eliminated on event $\mathcal{E}$. Note that since $FDR(\pi^*) < \alpha$

$$\widehat{FDR}(\pi^*) - \alpha \overset{\mathcal{E}}{\leq} FDR(\pi^*) - \alpha + C_1(\pi, t_k, \delta_k)/|\pi|$$
$$< C_1(\pi, t_k, \delta_k)/|\pi|.$$

Also for any $\pi \in \Pi_\alpha$,

$$\widehat{TP}(\pi \setminus \pi^*) - \widehat{TP}(\pi^* \setminus \pi) \overset{\mathcal{E}}{\leq} TP(\pi \setminus \pi^*) - TP(\pi^* \setminus \pi) + C_2(\pi, \pi^*, t_k, \delta_k)$$
$$= TP(\pi) - TP(\pi_*) + C_2(\pi, \pi^*, t_k, \delta_k)$$
$$\leq C_2(\pi, \pi^*, t_k, \delta_k),$$

and by definition $\pi^*$ is the maximal $TP$ set in $\Pi_\alpha$ so $\pi^*$ will never be removed by another $\pi$.

Finally note that on event $\mathcal{E}$, any $\pi'$ (not just $\pi_*$) can knock out $\pi$ using line 2 or 3 of the algorithm iff $TP(\pi') > TP(\pi)$ and $\pi' \in \Pi_\alpha$.

We define a few key random rounds

$$K_\pi := \max\{k : \pi \in \mathcal{A}_k\}$$
$$K_\pi^{FDR,1} := \max\{k : \pi \in \mathcal{A}_k \setminus \mathcal{C}_k\}$$
$$K_\pi^{FDR,2} := \min\{k : |\widehat{FDR}(\pi) - \alpha| > C_1(\pi, t_k, \delta_k)\}$$
$$K_\pi^{TP} := \min\{k : \exists \pi' \in \mathcal{C}_k \text{ such that } \widehat{TP}(\pi' \setminus \pi) - \widehat{TP}(\pi \setminus \pi') > C_2(\pi', \pi, t_k, \delta_k)\}$$
$$K_\pi^{<} := \min\{k : \exists \pi' \in \mathcal{C}_k \text{ with } \pi \subset \pi'\}$$

Our objective is to bound $\max_{\pi \in \Pi \setminus \pi_*} K_\pi$, which marks the termination of the algorithm.

**Bound on $K_\pi^{FDR,1}$:** We begin by establishing a deterministic bound on $K_\pi^{FDR,1}$ that holds when event $\mathcal{E}$ is true. Note that $K_\pi^{FDR,1}$ is immediately before the first $k$ such that $\pi \notin \mathcal{A}_k \setminus \mathcal{C}_k$. There are three ways this can occur: i) if $\pi$ becomes FDR-controlled or if $\pi$ is determined to not be FDR-controlled, and ii) a $\pi' \in C_k$ knocks out $\pi$ using statistics about $TP$ (i.e., line 2 of the algorithm), or iii) a $\pi' \in C_k$ knocks out $\pi$ deterministically by line 3 of the algorithm. These cases are reflected with the $\min$ respectively:

$$K_\pi^{FDR,1} = \min\{K_\pi^{FDR,2}, K_\pi^{TP}, K_\pi^{<}\}.$$

We provide a bound for each one of these terms under $\mathcal{E}$.

- Since $C_1(\pi, t_k, \delta_k)$ is a decreasing function of $k$, note that

$$|FDR(\pi) - \alpha| > 2C_1(\pi, t_k, \delta_k)/|\pi| \implies |\widehat{FDR}(\pi) - \alpha| > C_1(\pi, t_k, \delta_k)/|\pi|$$

  so on event $\mathcal{E}$, $K_\pi^{FDR,2} < k_\pi^{FDR,2}$ where

$$k_\pi^{FDR,2} := \min\{k : \Delta_{\pi,\alpha}/2 > C_1(\pi, t_k, \delta_k)/|\pi|\}.$$

- On event $\mathcal{E}$, only sets from $\Pi_\alpha$ will enter $\mathcal{C}_k$, so only they can be used to knock out other sets in Line 2 of the algorithm. Since $\pi^*$ is never eliminated on event $\mathcal{E}$, we have that:

$$K_\pi^{TP} \overset{\mathcal{E}}{\leq} \min\{k : \pi^* \in \mathcal{C}_k \text{ and } \widehat{TP}(\pi^* \setminus \pi) - \widehat{TP}(\pi \setminus \pi^*) > C_2(\pi^*, \pi, t_k, \delta_k)\}.$$

  Thus denoting $\Delta_\pi = TP(\pi^* \setminus \pi) - TP(\pi \setminus \pi^*)$ let

$$k_\pi^{TP} := \min\{k : \Delta_\pi/2 > C_2(\pi^*, \pi, t_k, \delta_k) \text{ and } \Delta_{\pi^*,\alpha}/2 > C_1(\pi^*, t_k, \delta_k)/|\pi^*|\}$$

  and note that $K_\pi^{TP} \overset{\mathcal{E}}{\leq} k_\pi^{TP}$ (note that this is potentially infinite if $TP(\pi) > TP(\pi_*)$).

- Using similar logic, on event $\mathcal{E}$ a set $\pi'$ will knock out a set $\pi$ using Line 3 of the algorithm only if $\pi'$ is in $\mathcal{C}_k \cup R$ and $\pi \subset \pi'$. If $\pi' \in \mathcal{C}_k$ then $TP(\pi') \geq TP(\pi)$ so we can remove $\pi$. If $\pi' \in R$ but $\pi' \notin \mathcal{C}_k$ yet, there exists a $\pi'' \in \mathcal{C}_k$ (in particular, the $\pi''$ that eliminated $\pi'$ into $R$) with $TP(\pi'') > TP(\pi') > TP(\pi)$ so we can safely remove $\pi$. Either way this implies that the $K_\pi^{<}$ is bounded by the time it takes to guarantee that $\pi'$ is FDR-controlled, hence

$$K_\pi^{<} \overset{\mathcal{E}}{\leq} \min_{\substack{\pi' \in \Pi_\alpha \\ \pi \subset \pi'}} K_{\pi'}^{FDR,2} \overset{\mathcal{E}}{\leq} \min_{\substack{\pi' \in \Pi_\alpha \\ \pi \subset \pi'}} k_{\pi'}^{FDR,2}.$$

Putting all of this together we set

$$k_\pi^{FDR,1} := \min\{k_\pi^{FDR,2}, k_\pi^{TP}, \min_{\substack{\pi \in \Pi_\alpha \\ \pi \subset \pi'}} k_{\pi'}^{FDR,2}\} \tag{4}$$

This is necessarily finite since $k_\pi^{FDR,2}$ is finite.

**Summarizing:**, on event $\mathcal{E}$, $k_\pi^{FDR,1}$ is an upper bound on $K_\pi^{FDR,1}$, the minimal round where $\pi \notin \mathcal{A}_{k+1} \setminus \mathcal{C}_{k+1}$.

**Part 2 Bound on $K_\pi$:** If $\pi \in \Pi_\alpha$, on event $\mathcal{E}$, $\pi$ will be removed from $\mathcal{A}_k$ only when it demonstrably has lower $TP$ than some other set $\pi' \in \Pi_\alpha$ regardless of whether it is in $\mathcal{C}_k$ or not. If $\pi \notin \Pi_\alpha$, on event $\mathcal{E}$, $K_\pi^{FDR,1} = K_\pi$, since the moment it's FDR is confirmed to be greater than $\alpha$ it is removed. Hence using the exact same logic as above, we have $K_\pi \overset{\mathcal{E}}{\leq} k_\pi$ where

$$k_\pi := \begin{cases} \min\{k_\pi^{TP}, \min_{\substack{\pi' \in \Pi_\alpha \\ \pi \subset \pi'}} k_{\pi'}^{FDR,2}\} & \pi \in \Pi_\alpha \\ k_\pi^{FDR,1} & \pi \notin \Pi_\alpha \end{cases} \tag{5}$$

**Summarizing:** On event $\mathcal{E}$, $k_\pi$ is an upper bound on $K_\pi$ and thus the algorithm terminates at some random round $K \leq k_{\max} := \max_{\pi \in \Pi \setminus \pi_*} k_\pi$ and outputs $\pi_*$.

**Part 3: Bound the contribution of each arm.** By the last step, we clearly have that the total sample complexity is bounded by

$$\sum_{k=1}^{k_{\max}} \sum_{t=t_{k-1}+1}^{t_k} \mathbf{1}\{I_t \in S_k\} + \mathbf{1}\{J_t \in T_k\}.$$

Since $I_t, J_t$ are uniformly distributed over $[n]$, we have $\mathbb{E}[\mathbf{1}\{I_t \in S_k\}|S_k] = \frac{|S_k|}{n}$ and $\mathbb{E}[\mathbf{1}\{J_t \in T_k\}|T_k] = \frac{|T_k|}{n}$. However, because $|S_k|$ and $|T_k|$ are random variables, we will upper bound them by deterministic quantities, and then show that the sample complexity concentrates.

For each $i \in [n]$, in round $k$, note that arm $i \in S_k$ if there is a set $\pi \in \mathcal{A}_k \setminus \mathcal{C}_k$ with $i \in \pi$. Hence

$$S_k = \{i \in [n] : \exists \pi \in \Pi : K_\pi^{FDR,1} > k\} \overset{\mathcal{E}}{\subset} \{i \in [n] : \exists \pi \in \Pi : k_\pi^{FDR,1} > k\} =: \psi_k$$

Similarly, $i \in T_k$ if there is $\pi, \pi' \in \mathcal{A}_k$ with $i \in \pi \Delta \pi'$. On event $\mathcal{E}$, $\pi^* \in \mathcal{A}_k$ for all $k$, thus $i \in T_k$ iff $i \in \pi \Delta \pi^*$ for some $\pi \in \mathcal{A}_k$. Thus

$$T_k = \{\pi \in \Pi : i \in \pi \Delta \pi^*, K_\pi > k\} \overset{\mathcal{E}}{\subset} \{\exists \pi \in \Pi : i \in \pi \Delta \pi^*, k_\pi > k\} =: \tau_k$$

We now follow an argument similar to that in the proof of Theorem 1. Thus $\mathbf{1}\{I_t \in S_k\} \leq \mathbf{1}\{I_t \in \psi_k\}$ and $\mathbf{1}\{J_t \in T_k\} \leq \mathbf{1}\{J_t \in \tau_k\}$ regardless of whether $I_t, J_t$ are drawn uniformly at random from $[n]$ or uniformly at random from $[n] \setminus \{i : I_s = i, 1 \leq s \leq t\}$ respectively $[n] \setminus \{i : J_s = i, 1 \leq s \leq t\}$. In particular, $I_t, J_t$ are negatively associated so we can apply standard multiplicative Chernoff Bounds. In particular,

$$\mathbb{P}\Big(\sum_{k=1}^{k_{\max}} \sum_{t=t_{k-1}+1}^{t_k} \mathbf{1}\{I_t \in S_k\} \geq (1+r) \sum_{k=1}^{k_{\max}} t_k \frac{|\psi_k|}{n}\Big)$$

$$\leq \mathbb{P}\left(\sum_{k=1}^{k_{\max}} \sum_{t=t_{k-1}+1}^{t_k} \mathbf{1}\{I_t \in \psi_k\} \geq (1+r) \sum_{k=1}^{k_{\max}} t_k \frac{|\psi_k|}{n}\right)$$

$$\leq \exp\left(-\frac{\min\{r, r^2\}}{3} \sum_{k=1}^{k_{\max}} t_k \frac{|\psi_k|}{n}\right)$$

577 with the appropriate choice of $r$, with probability greater than $1 - \delta$,

$$\sum_{k=1}^{k_{\max}} \sum_{t=t_{k-1}+1}^{t_k} \mathbf{1}\{I_t \in S_k\} \le \frac{9}{2} \log(2/\delta) + \frac{3}{2} \sum_{k=1}^{\infty} t_k \frac{|\psi_k|}{n}$$

578 An identical argument gives that with probability greater than $1 - \delta$,

$$\sum_{k=1}^{k_{\max}} \sum_{t=t_{k-1}+1}^{t_k} \mathbf{1}\{J_t \in T_k\} \le \frac{9}{2} \log(2/\delta) + \frac{3}{2} \sum_{k=1}^{\infty} t_k \frac{|\tau_k|}{n}.$$

579 While we have provided a bound on the sample complexity in terms of deterministic quantities $\psi_k$
580 and $\tau_k$, we now want to provide natural and interpretable upper bounds on these quantities for a final
581 result.

582 Putting it all together we have that

$$
\begin{aligned}
\sum_{k=1}^{\infty} t_k \frac{\psi_k + \tau_k}{n} &= \sum_{k=1}^{\infty} \frac{2^k}{n} (\psi_k + \tau_k) \\
&= \sum_{i=1}^{n} \sum_{k=1}^{\infty} \frac{2^k}{n} (\mathbf{1}\{\exists \pi \in \Pi : i \in \pi, k_\pi^{FDR,1} > k\} \\
&\qquad + \mathbf{1}\{\exists \pi \in \Pi : i \in \pi \Delta \pi^*, k_\pi > k\}) \\
&\le \sum_{i=1}^{n} \sum_{k=1}^{\infty} \frac{2^k}{n} (\mathbf{1}\{\exists \pi \in \Pi : i \in \pi, k_\pi^{FDR,1} > k\} \\
&\qquad + \mathbf{1}\{\exists \pi \in \Pi, \pi \in \Pi_\alpha : i \in \pi \Delta \pi^*, k_\pi > k\} \\
&\qquad + \mathbf{1}\{\exists \pi \in \Pi, \pi \notin \Pi_\alpha : i \in \pi \Delta \pi^*, k_\pi > k\}) \\
&\le \sum_{i=1}^{n} \max_{i \in \pi} \frac{2^{k_\pi^{FDR,1}+1}}{n} + \max_{\substack{\pi \notin \Pi_\alpha \\ i \in \pi \Delta \pi^*}} \frac{2^{k_\pi^{FDR,1}+1}}{n} + \max_{\substack{\pi \in \Pi_\alpha \\ i \in \pi \Delta \pi^*}} \frac{2^{k_\pi+1}}{n} \\
&\le \sum_{i=1}^{n} 2 \max_{i \in \pi} \frac{2^{k_\pi^{FDR,1}+1}}{n} + \max_{\substack{\pi \in \Pi_\alpha \\ i \in \pi \Delta \pi^*}} \frac{2^{k_\pi+1}}{n}
\end{aligned}
$$

583 The fourth line follows from Equation (5) and the last line follows from upper bounding the second
584 term in the fourth line by the first. Solving for $k$, shows that for some constant $c_1$

$$
\begin{aligned}
2^{k_\pi^{FDR,2}} &\le \min \left\{ m : 2C(\pi, n, m, \delta_{\lfloor \log_2(m) \rfloor}) < |FDR(\pi) - \alpha| \right\} \\
&\le c_1 n V_\pi \frac{\log(n \log(\Delta_{\pi,\alpha}^{-2})}{|\pi| \Delta_{\pi,\alpha}^2}
\end{aligned}
$$

585 An identical argument shows that for arbitrary $\pi, \pi'$, there is a constant $c_2$ such that

$$
\begin{aligned}
2^{k_\pi^{TP}} &\le \max \left\{ c_2 n V_{\pi,\pi^*} \left( \frac{|\pi \Delta \pi^*|}{\Delta_\pi^2} + \frac{1}{\Delta_\pi} \right) \log \left( \frac{n \log(\Delta_\pi^{-2})}{\delta} \right), 2^{k_{\pi^*}^{FDR,2}} \right\} \\
&= \max \left\{ c_2 \frac{n V_{\pi,\pi^*}}{|\pi \Delta \pi^*|} \frac{1}{\widetilde{\Delta}_\pi^2} \log \left( \frac{n \log(\widetilde{\Delta}_\pi^{-2})}{\delta} \right), 2^{k_{\pi^*}^{FDR,2}} \right\}
\end{aligned}
$$

586 Finally, for the persistent noise case we have $k_\pi, k^{FDR,2} \le \log_2(n)$ which implies for any $i$,
587 $\max_{\pi \in \Pi : i \in \pi_* \Delta \pi} \frac{2^{k_\pi+1}}{n} \le 2$. The theorem now follows.

$\square$

588

# C  One-dimensional thresholds

We can get tighter characterizations of Lemma  and consequently, better sample complexity guarantees for particular VC classes. In particular, those classes that have sets with substantial overlap like thresholds. In the case of **Thresholds** we have the following improvement that manages to remove the extra $\log(n)$ terms in Lemma 1.

**Lemma 2** *Assume that for each $i \in [n]$ there is an associated distribution $\nu_i$ with support $[-1, 1]$, mean $\mu_i$ and variance $\sigma_i^2 \leq 1$. Assume access to the observations $(y_1, I_1)\cdots, (y_T, I_T)$ where $I_k \sim Unif([n])$ and $y_k \sim \nu_{I_k}$. Let $\widehat{\mu}_t = \frac{1}{T}\sum_{k=1}^{T} y_k \mathbf{1}\{I_k \leq t\}$. Fix $t' \leq n$. Then with probability greater than $1 - \delta$ for any $s \leq n$,*

$$|\widehat{\mu}_s - \widehat{\mu}_{t'} - (\mu_s - \mu_{t'})| \leq \sqrt{\tfrac{2|s-t'|}{nT}} \left(43 + 2\sqrt{2}\log(2\log_2^2(4|s-t'|)/3\delta))\right) + \tfrac{12+\log\left(2\log_2^2(4|s-t'|)/3\delta\right)}{3T}$$

An analogous result can be proven in the persistent noise case of sampling without replacement.

**Active Classification for One-dimensional thresholds with Tsybakov Noise** - Let $h \in (0,1]$, $\alpha \geq 0$, $z \in [0,1]$ for some $i \in [n-1]$ and assume that $X_{i,j} \in \{-1,1\}$ are Bernoulli with $\mathbb{P}(X_{i,j} = \text{SIGN}(z - i/n)) = \frac{1}{2} + \frac{1}{2}h|z - i/n|^\alpha$ so that $\mu_i = h|z - i/n|^\alpha \text{SIGN}(z - i/n)$. Let $\Pi = \{[k] : k \leq n\}$. In this case, inspecting the dominating term of 1 for $i \in \pi^*$ we have $\arg\max_{\pi\in\Pi: i\in\pi\delta\pi^*} \frac{V_{\pi,\pi^*}}{|\pi\Delta\pi^*|}\frac{1}{\Delta_\pi^2} = [i]$ and takes a value of $\left(\frac{1+\alpha}{h}\right)^2 n^{-1}(z - i/n)^{-2\alpha-1}$. Trivially upper bounding the other terms and summing, the sample complexities can be calculated to be within a constant of

$$\text{if } \alpha = 0, \ \log(n)\log(\log(n)/\delta)/h^2 \qquad \text{if } \alpha > 0 \quad n^{2\alpha}\log(\log(n)/\delta)/h^2$$

These rates match the minimax lower bound rates given in [12] up to $\log\log$ factors. Note that unlike the algorithms given there, our algorithm works in the *agnostic* setting, i.e. it is making no assumptions about whether the Bayes classifier is in the class. In the case of non-adaptive sampling, the sum is replaced with the max times $n$ yielding

$$\text{if } \alpha \geq 0 \ n^{2\alpha+1}\log(\log(n)/\delta)/h^2$$

which is substantially worse than adaptive sampling.

We are now ready to prove the theorem.

**Proof:**  Let

$$f_t(I_k, y_k) = \begin{cases} y_k \mathbf{1}\{I_k \in [t', t]\} & t \geq t' \\ -y_k \mathbf{1}\{I_k \in [t, t']\} & t \leq t' \end{cases}$$

In particular, $\widehat{\mu}_t - \widehat{\mu}_{t'} = \frac{1}{T}\sum_{k=1}^{T} f_t(I_k, y_k)$. Note that the random variables $(y_s, I_s)$, for $s = 1, \cdots, n$ are by definition i.i.d. drawn from a distribution on $[n] \times \{0, 1\}$. Note

$$\mathbb{E}\left[\frac{1}{T}\sum_{k=1}^{n} f_t(I_k, y_k)\right] = \begin{cases} \frac{1}{n}\sum_{k=t'}^{t} \eta_i & t \geq t' \\ \frac{1}{n}\sum_{k=t}^{t'} -\eta_i & t \leq t' \end{cases}$$

and (assuming that $t \leq t'$, an identical computation applies when $t \geq t'$)

$$\begin{aligned} \text{var}(f_t) &= \text{var}(y_s \mathbf{1}\{I_s \in [t, t']\}) \\ &\leq \mathbb{E}[y_s^2 \mathbf{1}\{I_s \in [t, t']\}] \\ &= \frac{1}{n}\sum_{i=t}^{t'}(\sigma_i^2 + \eta_i^2) \leq \frac{2}{n}|t' - t|. \end{aligned}$$

By Theorem 2.3 in [7], given $\delta > 0$, for each $\{s : s \leq n, |s - t'| \leq \tau\}$ we have that

$$\mathbb{P}\Bigg( \left|\frac{1}{T}\sum_{k=1}^{T} f_s(I_k, y_k) - \mathbb{E}[f_s]\right| > 2\mathbb{E}\left[\sup_{|s-t'|\leq\tau} \left|\frac{1}{T}\sum_{k=1}^{T} f_s(I_k, y_k) - \mathbb{E}[f_s]\right|\right]$$

$$+ \sqrt{\frac{2\tau\log(1/\delta)}{nT}} + \tfrac{7\log(1/\delta)}{3T}\Bigg) \leq \delta$$

To obtain a bound over all time, we now face two major tasks. Firstly, we must apply a peeling argument to the set of $t$'s. Secondly, and perhaps more immediate, we need bounds on the empirical process

$$\mathbb{E}\left[\sup_{|s-t'|\leq\tau}\left|\frac{1}{T}\sum_{k=1}^{T}f_s(I_k,y_k)-\mathbb{E}[f_s]\right|\right]$$

Let's start with the latter. Denote $Z_t = \frac{1}{T}\sum_{k=1}^{T}f_t(I_k,y_k)-\mathbb{E}[\frac{1}{T}\sum_{k=1}^{T}f_t(I_k,y_k)]$. Firstly note that,

$$|(f_s-f_t)(I_k,y_k)| = \begin{cases} y_k\mathbf{1}\{I_k\in[t,s]\} & s>t \\ -y_k\mathbf{1}\{I_k\in[s,t]\} & t>s \end{cases}$$

In particular the computation above shows,

$$\mathrm{var}\left((f_s-f_t)(I_k,y_k)\right)\leq 2\frac{|t-s|}{n}.$$

Hence,

$$\mathrm{var}\left(\frac{1}{T}\sum_{k=1}^{T}f_t(I_k,y_k)-\mathbb{E}[f_t]-\left(\frac{1}{T}\sum_{k=1}^{T}f_s(I_k,y_k)-\mathbb{E}[f_s]\right)\right) = \frac{\mathrm{var}(f_t(I_k,y_k)-f_s(I_k,y_k))}{T}$$

$$\leq \frac{2|t-s|}{nT}$$

In particular, since $|\frac{1}{T}f_t(I_s,y_s)|\leq\frac{1}{T}$, Bernstein's inequality implies,

$$\log(\mathbb{E}[e^{\lambda(Z_t-Z_s)}])\leq\frac{\lambda^2\frac{2|t-s|}{nT}}{2(1-\lambda/3T)}.$$

Let $d^2(t,s) = |\frac{t}{n}-\frac{s}{n}|$. Then, Lemma 13.1 of [6] with $\nu = 2/T$ and $c = 1/3T$ we have that,

$$\mathbb{E}\left[\sup_{|s-t'|\leq\tau}|Z_s|\right]\leq\frac{12\sqrt{2}}{\sqrt{T}}\int_0^{\sqrt{\tau/n}/2}\sqrt{\log(\frac{\sqrt{\tau/n}}{2u})}du + \frac{4}{T}\int_0^{\sqrt{\tau/n}/2}\log(\frac{\sqrt{\tau/n}}{2u})du$$

$$\leq\frac{12\sqrt{2}}{\sqrt{T}}\int_0^{\infty}\sqrt{\frac{\tau}{n}}v^2e^{-v^2}dv + \frac{4}{T}\int_0^{\infty}\frac{1}{2}\sqrt{\frac{\tau}{n}}ve^{-v}dv$$

$$\leq\frac{12\sqrt{\pi}}{\sqrt{T}}\sqrt{\frac{\tau}{n}} + \frac{2}{T}\sqrt{\frac{\tau}{n}}$$

$$\leq 12\sqrt{\pi}\sqrt{\frac{\tau}{nT}} + \frac{2}{T}$$

the third line follows from the second by doing the substitution, $v = \sqrt{\log(\sqrt{\tau/n}/u)}$ and similarly $u = \log(\sqrt{\tau/n}/u)$ on the second integral.

Hence for all $s : |s-t'|\leq\tau$, using the fact that $\sqrt{a}+\sqrt{b}\leq\sqrt{2(a+b)}$

$$\mathbb{P}\left(\left|\frac{1}{T}\sum_{k=1}^{T}f_s(I_k,y_k)-\mathbb{E}[f_s]\right| > \sqrt{\frac{\tau}{nT}\left(43+2\sqrt{2}\log(\frac{1}{\delta})\right)} + \frac{12+\log(1/\delta)}{3T}\right)\leq\delta$$

At this point we need to apply a peeling argument. Let $S_r = \{s\leq n : 2^{r-1}\leq|s-t'|\leq 2^r\}$. Note that $r\leq\log_2(2|s-t'|+2)\leq\log_2(4|s-t'|)$. For each $s\in S_r$ simultaneously, since $2^r\leq 2|s-t'|$, with probability greater than $1-\frac{2\delta}{3r^2}$,

$$\left|\frac{1}{T}\sum_{k=1}^{T}f_s(I_k,y_k)-\mathbb{E}[f_s]\right| < \sqrt{\frac{2|s-t'|}{nT}\left(43+2\sqrt{2}\log(\frac{2r^2}{3\delta})\right)} + \frac{12+\log\left(\frac{2r^2}{3\delta}\right)}{3T}$$

$$\leq\sqrt{\frac{2|s-t'|}{nT}\left(43+2\sqrt{2}\log(\frac{2\log_2^2(4|s-t'|)}{3\delta})\right)} + \frac{12+\log\left(\frac{2\log_2^2(4|s-t'|)}{3\delta}\right)}{3T}$$

632  Now union-bounding over each $r = 1, \cdots, \log_2(n - t')$, we have that

$$\left| \frac{1}{T} \sum_{k=1}^{T} f_s(I_k, y_k) - \mathbb{E}[f_s] \right| \leq \sqrt{\frac{2|s-t'|}{nT} \left( 43 + 2\sqrt{2} \log\left(\frac{2\log_2^2(4|s-t'|)}{3\delta}\right) \right)} + \frac{12 + \log\left(\frac{2\log_2^2(4|s-t'|)}{3\delta}\right)}{3T}$$

633  with probability greater than

$$\sum_{k=1}^{\log_2(n-t')} \frac{2\delta}{3k^2} \leq \sum_{k=1}^{\infty} \frac{2\delta}{3} k^2 \leq \delta$$

634                                                                                           □