[Reviews · NeurIPS 2019]

Reviewer 1



This manuscript considers a novel active learning criterion: namely one of maximizing the true positive rate (TPR) subject to a constraint on the false discovery rate (FDR). The authors draw connections between this paradigm and several existing lines of work including combinatorial bandits and active classification. The technical results appear sound and interesting. The authors show sample complexity results for FDR control and for pool-based active learning in terms of novel complexity measures. The paper is predominantly clearly written. However, I have some minor comments that could aid in enhancing the quality of the presentation: - The notation, especially in the statement of Theorem 2, is quite difficult to parse. I would encourage the authors to consider ways to aid readability. - Please provide a sketch of the proof in the main text that highlights the main techniques and difficulties. I think this manuscript addresses an interesting problem and the presents novel theoretical results and techniques; I think this paper will make a great addition to NeurIPS 19.

Reviewer 2



Originality: The problem considered in this paper has not been extensively studied yet. The proposed solution is based on a nice combination of techniques from active learning and combinatorial bandits. Quality: I didn't check proofs in appendix, but results look reasonable to me. Clarity: This paper is well-organized. However, its technical part is a little bit dense and more explanation might be helpful. Below are some detailed comments: 1. It is very nice to motivate the problem with an application in the introduction. However, the example given is a little confusing to me. For example, Figure 1 is not well explained (what's the difference between babb and aaa? what does the distribution of bueired NPSA mean? What do you mean by "the distribution of a feature that is highly correlated with the fitted logistic model"?). These details might be irrelevant, but can be distracting for readers. Another issue is that when I was reading this part for the first time, at some point I thought the main issue was sampling bias or class imbalance, but actually this is not the point the authors want to make. 2. It might be easier to read if the authors could explain some high level intuitions for Algorithms 1 and 2 before explaining details. 3. Many definitions are not well explained (for example, V_\pi, V_{\pi, \pi'} in line 178-179, \tau_\pi in line 184-185, ...). Explaining these might shed some lights on how the algorithm/proof works and how much improvement had been made. 4. There seem to be some contributions this paper claims but are not very well explained in the main body. For example, the log(n) factors mentioned in line 121, local VC dimensions in line 180, the improvement over disagreement coefficient in section 3.1. Significance: This problem is well-motivated, and results are better than standard baselines/existing methods. One downside though is that the techniques seem a straightforward extension of existing ones. == After rebuttal: The authors have clarified some of my concerns. I hope the authors could improve the clarity of the technical part, and be more specific about the main challenges and the significance of its contribution in the future version.

Reviewer 3



First I want to say that I like the problem setting and the general approach. Also I do not have a single main concern, but I have several non-negligible ones that, when piled up, determined my final score. In what follows, I’ll order my concerns/questions from more important ones to minor comments. I felt that the main result, which is given in Theorem 2, was really hard to interpret. There are way too many terms, making their combination in the final result very hard to interpret. I found the discussion following the theorem very helpful, but it helped me understand how these terms occurred in the proof, rather than what their practical meaning is. I am happy seeing such a theorem if it is followed by multiple corollaries, given by different parameter instantiations, showing when the bound is tight and how the obtained complexity compares to prior work. Some comparison with the result of [4] is given, although in the fully general setting; it would be interesting to see settings when this difference is significant, and when it is not. For example, I liked the “One Dimensional Thresholds” example. Perhaps this is a matter of taste, but when I reached the last paragraph, I thought the paper ended too early; I was expecting more such examples to follow. 
Another concern is that there have been multiple works on different bandit approaches to multiple testing (not all of which test for whether the mean of a distribution is zero, as stated in the paper). Some related papers that weren’t discussed include, for example, Optimal Testing in the Experiment-rich Regime by Schmit et al. (AISTATS 2019), A framework for Multi-A(rmed)/B(andit) testing with online FDR control by Yang et al. (NeurIPS 2017), etc. Moreover, these papers focus on sample complexity just like the current paper. Their setting is different, but the papers are similar enough that they deserve a brief discussion. Citing only 3 papers in the “Multiple Hypothesis Testing” paragraph makes me think that the connections to prior work in this area have not been fully explored. If I understand correctly, the proposed algorithm is inspired by algorithms in related work (e.g. [4, 10]), but a fundamentally different idea is sampling in the symmetric difference, which has not been exploited so far? Further, I found the writing very messy in some parts, and it took a lot of re-reading to go over some definitions and arguments. For example, A_k and T_k are introduced without stating what k is. At the end of page 4, there is a sentence that says “the number of samples that land in T_k”, even though T_k is not a set of samples. In the same paragraph, mu-hats are used without being previously defined. The paragraph assumes I_t is uniformly distributed on [n], even though that is not stated beforehand. In the definition of TPR, eta_[n] is used, even though it is never defined. I understand that submissions are thoroughly revised before publication, but these kinds of mistakes make reviewing harder than it needs to be. I don’t know if this is necessary for the proof or not, but I didn’t see why the mu-hat difference is unbiased, as stated at the bottom of page 4. Especially if the formal result hinges on this fact, I would appreciate a formal argument explaining why this is the case. The rejection sampling strategy is essentially equivalent to the following: conditioned on “past information”, sample uniformly from the set T_k. This couples the old and new samples, making it not so obvious to me that the mu-hat difference is unbiased. Related to this point, I didn’t understand the part that says that the number of samples that land in T_k follow a geometric distribution. I agree that the wait time until you observe a selection in T_k is a geometric random variable. Relatively minor comments regarding style: 1. It is incorrect to say that Y_{I_t,t} is iid, as written at the beginning of page 3; iid is an assumption that refers to a set of random variables. This sentence needs to be rephrased more formally. 2. I was confused by the sentence “Instead of considering all possible subsets of [n], we will restrict ourselves to a finite class … .” The set of all subsets of [n] is finite anyway. 3. There is a minus sign missing when introducing mu_i in the risk equation on page 4. Either way, I do not see the purpose of introducing the risk. The paper makes it clear that the goal is to maximize TPR given an FDR constraint, as opposed to minimizing risk. Much of Section 1.1. seems like an unnecessary distraction. 4. There are some typos that need revision, like at the beginning of Section 3.1 where it says “specific specific noise models”, or in the Remark on page 5 there should be R_{i,t}. 5. The Benjamini-Hochberg paper should probably be cited, given that the used error metric is FDR, which stems from that paper. After rebuttal: The authors have clarified some of my concerns, and have promised to improve clarity of presentation. The other reviews have also convinced me about the originality of this submission, so I am increasing my score.

[Author Response · NeurIPS 2019]

Thanks to all reviewers for their extensive feedback. The goal of this paper was two-fold, we proposed a new and important problem setting of active learning for combinatorial pool-based FDR control - a problem of tantamount importance in the sciences that has received little to no attention and gave a state of the art algorithm and analysis for pool-based active classification that achieves optimal bounds in several important settings. Achieving this required several technical advancements that show structural connections between bandits and active learning that has so far gone unnoticed in the machine learning community. We are excited to share these contributions and ideas with the NeurIPS community. We now dive into these ideas more carefully and address specific comments by the reviewers.

**Reviewer #1:** Thank you for the encouraging review. We will add a simplified statement of the theorem, which while not being optimal will illustrate the two key terms - the sample complexity of FDR verification combined with demonstrating the highest TPR set. The proof relies heavily on the sampling scheme and the choice of estimators. We will expand on this proof in the main body.

**Reviewer #3:** Thank you for your comments. We first address your concerns about significance. Utilizing a bandit-based analysis for active classification is a new strategy that promises to lead to new breakthroughs in active learning algorithms. In a recent work ([10]), the authors explicitly state that an pure exploration elimination strategy doesn't seem to address problems in active learning - a hurdle we overcome (partially due to not needing to sample each arm once) while giving improved sample complexity bounds in this setting. Vice versa, empirical process techniques could lead to new methods of dealing with union bounds in combinatorial bandit based algorithms.

This also relates to your concerns in points 3 and 4. Bounds in statistical learning theory based on VC (local) dimensions are well-understood and standard in the passive case to replace finite class union bounds with bounds over infinite classes. They have received less attention in the bandit and active learning literature. The expression given in line 178 and the result of Theorem 1 are novel contributiona and are motivated more thoroughly in the appendix (see A.1) and in the form of the confidence bound given there. We will move a summary of this discussion into the main body.

In the introduction we will clarify the plot adding a sentence like "... a particularly informative feature (Buried NPSA) are shown in each round for two different protein topologies (notated $\beta\alpha\beta\beta$, and $\alpha\alpha\alpha$)". We motivated algorithm 1 in lines 159-175, and algorithm 2 in lines 233-252 along with Figure 2. We will provide additional motivation emphasizing the algorithm simultaneously samples to guarantee FDR-control while removing sets with sub-optimal TPR.

**Reviewer #4:** Thank you for thorough review and comments. We understand that Theorem 2 can be hard to unpack, but we hoped the discussion in lines 267-282 along with Figure 2 would clarify it - a discussion we are happy to expand. We will provide additional settings where our algorithm will perform significantly better than uniform sampling and [4]. This includes cases when there are a large number of sets that can be FDR-controlled quickly, but then are eliminated based on TPR - a setting in which our confidence intervals are much tighter than those given in [4].

Regarding multiple testing, we explain in the related works section (lines 145-157) that this paper (though it borrows terminology from that domain) is not at all related to multiple testing. We are familiar with the works you referenced - and made a deliberate choice to not discuss them further, though we agree that we should cite them. The multiple testing setting has a different goal and less structure from ours - the goal there is to find which individuals $i \in [n]$ have means $\mu_i$ above a threshold. In our setting we are seeking to return a hypothesis in a fixed class that has high TPR under an FDR constraint.

We appreciate your concerns about the writing and will clarify the algorithm further. In the discussion before Algorithm 1, we were assuming that the reader was reading the discussion in 164- 175 simultaneously while viewing the algorithm display - we will signpost this better. Sampling in the symmetric difference is a strategy that has appeared before in [4,10], however, building estimators that do not require each item to be sampled once, work in an agnostic noise setting, an analysis based on gaps, and connections to VC dimension are all improvements on these works.

The estimator $\widehat{\mu}_{\pi',k} - \widehat{\mu}_{\pi,k} = \frac{n}{t} \sum_{s=1}^{t} R_{I_t,s}(\mathbf{1}(I_s \in \pi' \setminus \pi) - \mathbf{1}(I_s \in \pi \setminus \pi'))$ depends on *all* $t$ samples up to the $t$-th rounds, each of which is uniformly and independently drawn at each step. Thus each summand is an unbiased estimate of $\mu_{\pi'} - \mu_\pi$. However, for $\pi, \pi'$ active in round $k$, a summand is only non-zero if $I_s \in \pi \Delta \pi' \subset T_k$ hence we only need to observe $R_{I_t,s}$ if $I_t \in T_k$. This is equivalent to the rejection sampling procedure given: take a sample from an appropriate binomial distribution (not a geometric as pointed out) and then drawing that many indices from $T_k$ and observing rewards from their associated distributions. We will say this more formally in the paper.

Regarding point 3 in the stylistic comments. Thank you for the math typo, indeed a minus sign is missing. Section 1.1 motivates active classification which we address in Section 3 with the goal of motivating sampling in the symmetric difference. This is an idea that is critical in Algorithm 2 for determining the set with the highest TPR. Addressing other style comments:1. We will change this to say something like"$Y_{I_t,t}$ is an independent R.V. For any $i$, $Y_{i,t}$ for all $t$ are iid". 2. By "finite class", we meant "smaller class". 4. Thank you for the typos. 5. We mentioned Benjamini-Hochberg in the related works, and agree that citing the original work for this terminology is warranted.

[Meta-Review · NeurIPS 2019]

As the reviewers agree, this paper gives a nice connection between active learning and combinatorial bandits, and provides concrete performance guarantees that outperform prior approaches. The reviewers would like to ask the authors to improve clarity in the final version of the paper.